# Predicting mutational effects on protein-protein binding via a side-chain diffusion probabilistic model

**Shiwei Liu**[1,2*]     **Tian Zhu**[1,2*]     **Milong Ren**[1,2]     **Chungong Yu**[1,2]

**Dongbo Bu**[1,2,3]                    **Haicang Zhang**[1,2†]

1. Institute of Computing Technology, Chinese Academy of Sciences, Beijing 100080, China.
2. University of Chinese Academy of Sciences, Beijing 100080, China.
3. Central China Research Institute for Artificial Intelligence Technologies, Zhengzhou 450046, China

## Abstract

Many crucial biological processes rely on networks of protein-protein interactions. Predicting the effect of amino acid mutations on protein-protein binding is vital in protein engineering and therapeutic discovery. However, the scarcity of annotated experimental data on binding energy poses a significant challenge for developing computational approaches, particularly deep learning-based methods. In this work, we propose SidechainDiff, a representation learning-based approach that leverages unlabelled experimental protein structures. SidechainDiff utilizes a Riemannian diffusion model to learn the generative process of side-chain conformations and can also give the structural context representations of mutations on the protein-protein interface. Leveraging the learned representations, we achieve state-of-the-art performance in predicting the mutational effects on protein-protein binding. Furthermore, SidechainDiff is the first diffusion-based generative model for side-chains, distinguishing it from prior efforts that have predominantly focused on generating protein backbone structures.

## 1 Introduction

Many crucial biological processes, including membrane transport and cell signaling, are likely regulated by intricate networks of protein-protein interactions rather than single proteins acting independently [Calakos et al., 1994, Jones and Thornton, 1996, Levskaya et al., 2009, Schwikowski et al., 2000]. One representative example is the interaction between the spike protein of the SARS-CoV-2 virus and its receptor protein ACE2 on the surface of human cells, which is crucial for the virus to invade target cells [Lan et al., 2020]. Meanwhile, specific human antibodies, an essential type of protein in the immune system, can prevent the virus entry by competitively binding to the spike protein [Shan et al., 2022].

In protein engineering and therapeutic discovery, it is a crucial step to induce amino acid mutations on the protein-protein interface to modulate binding affinity [T O'Neil and Hoess, 1995]. For example, to enhance the efficacy of an antibody against a virus, one practical approach is to introduce amino acid mutations and filter the resulting antibody sequences to increase binding affinity and specificity to the target viral protein [Mason et al., 2021]. However, the variant space that can be explored

---

[*]These authors contributed equally to this work.
[†]Correspondence should be addressed to H.Z. (zhanghaicang@ict.ac.cn)

37th Conference on Neural Information Processing Systems (NeurIPS 2023).

using experimental assays is very limited, and developing an effective high-throughput screening can require a significant experimental effort.

Computational methods have been developed to predict the mutational effect on binding affinity measured by *the change in binding free energy* (i.e., $\Delta\Delta G$). Traditional computational approaches mostly used physical energy features, such as van der Waals and solvation energy, in combination with statistical models to predict $\Delta\Delta G$ [Schymkowitz et al., 2005, Meireles et al., 2010, Alford et al., 2017, Barlow et al., 2018]. The limited model capacity and bias inherent in human-engineered features in these methods hinder their ability to characterize the complex mutational landscape of binding energy. Despite recent breakthroughs in protein modeling with deep learning [Jumper et al., 2021, Baek et al., 2021, Watson et al., 2023], developing deep learning-based models to predict mutational effects on protein-protein binding remains challenging due to the scarcity of labeled experimental data [Jankauskaitė et al., 2019].

Recent studies have investigated various self-supervised learning strategies on protein structures and sequences [Liu et al., 2021, Meier et al., 2021, Hsu et al., 2022, Luo et al., 2023] to ease the data scarcity issue. Among them, GeoPPI [Liu et al., 2021] and RDE [Luo et al., 2023] have focused on optimizing protein side-chain conformations during pre-training, as the side-chain conformations on the protein-protein interface play a critical role in determining binding energy. In protein-protein interactions, the side-chain conformations of amino acids at the interface may exhibit significant variations when comparing the same receptor with different ligands. These side-chain conformations can be more effectively characterized using probability density. Notably, RDE explores a flow-based model to estimate the uncertainty in side-chain conformations and leverages the learned representations to achieve state-of-the-art performance in predicting $\Delta\Delta G$. However, flow-based models possess inherent limitations as they require specialized architectures to construct accurate bijective transformations in probability density [Papamakarios et al., 2021], which results in increased costs associated with model design and implementation.

To address the above limitations, we propose SidechainDiff, a representation learning framework for the protein-protein interfaces. It employs a Riemannian diffusion model to learn the generative process of side-chain conformations and the representation of the structural context of amino acids. To the best of our knowledge, SidechainDiff is the first diffusion probabilistic model for side-chain modeling, whereas previous methods have only focused on protein backbone structures. Furthermore, we leverage the learned representations and neural networks to predict $\Delta\Delta G$.

## 2 Related Work

### 2.1 Protein side-chain conformation prediction

Accurate side-chain modeling is essential in understanding the biological functions of proteins. There are two primary categories in predicting protein side-chain conformations: end-to-end full-atom structure prediction methods and side-chain packing. AlphaFold and RoseTTAFold [Jumper et al., 2021, Baek et al., 2021] are two representative methods that simultaneously generate side-chain conformations and backbone structures. In scenarios like structure refinement and protein design, side-packing becomes pivotal. The objective is to determine the conformations of protein side-chains while having their backbone structures already defined. Traditional methods, including Rosetta [Leman et al., 2020] and SCWRL4 [Krivov et al., 2009] operate by minimizing the energy function across a pre-defined rotamer library. Recent methods for side-chain packing often employ deep learning models, such as 3D convolution networks and graph attention networks [McPartlon and Xu, 2023, Misiura et al., 2022, Xu et al., 2022].

Our model distinguishes itself from previous methods for side-chain modelling in two key aspects. First, it's capable of generating a distribution of side-chain conformations rather than a single conformation. Second, we emphasize side-chain modeling specifically for mutation sites within the protein-protein interface, leveraging the known structural context of these sites.

### 2.2 Prediction of mutational effects on protein-protein binding

Methods for predicting $\Delta\Delta G$ can be broadly classified into three categories: biophysical methods, statistical methods, and deep learning-based approaches. Biophysical methods offer a robust means of elucidating the molecular mechanisms governing protein-protein interactions and the impact of

mutations on these interactions [Schymkowitz et al., 2005, Alford et al., 2017, Steinbrecher et al., 2017]. These methods directly integrate information on protein structures and key biophysical properties, such as solvent accessibility, electrostatic potential, and hydrogen bonding patterns. Statistical methods tailor statistical models for the protein properties such as evolutionary conservation and geometric characteristics [Li et al., 2016, Geng et al., 2019, Zhang et al., 2020].

Deep learning-based methods can be categorized into sequence-based methods and structure-based methods. Sequence-based methods primarily either focus on the evolutionary history, multiple sequence alignments (MSAs) in most cases, of the target protein [Hopf et al., 2017, Riesselman et al., 2018, Frazer et al., 2021] or act as protein language models (PLMs) trained on large amounts of protein sequences [Meier et al., 2021, Notin et al., 2022]. Structure-based methods can be categorized into end-to-end methods and pre-training-based methods. The end-to-end methods extract features from protein complexes and directly train a neural network model on them [Shan et al., 2022]. To mitigate overfitting caused by data sparsity, an alternative approach is to learn representations by pre-training a feature extraction network on unlabeled structures [Liu et al., 2021, Luo et al., 2023]. Among them, RDE-Network [Luo et al., 2023] utilizes normalizing flows in torus space to estimate the density of amino acid side-chain conformations and leverages the learned representation to predict $\Delta\Delta G$.

## 2.3 Diffusion probabilistic models

Diffusion Probabilistic Models (DPMs) are generative models to transform a sample from a tractable noise distribution towards a desired data distribution with a gradual denoising process [Sohl-Dickstein et al., 2015, Kingma et al., 2021, Dhariwal and Nichol, 2021]. DPMs have achieved impressive results in generating various data types, including images [Ho et al., 2020, Nichol and Dhariwal, 2021], waveforms, and discrete data like text [Hoogeboom et al., 2021]. DPMs-based autoencoders have also been proposed to facilitate representation learning for image data [Preechakul et al., 2022, Zhang et al., 2022].

Inspired by these progresses, DPMs have also been explored in protein modeling, including de novo protein design [Watson et al., 2023, Ingraham et al., Yim et al., 2023], motif-scaffolding [Trippe et al., 2023], and molecular dynamics [Arts et al., 2023, Wu and Li, 2023]. While the existing methods utilize DPMs for generating protein backbones, a research gap remains in modeling side-chain conformations, which play a critical role in protein-protein binding. The studies by Jing et al. [2022] and Corso et al. [2023] employ diffusion models for generating torsion angles in the context of small molecular design. In contrast, our research is centered on modeling the torsional angles of protein side-chains. Furthermore, our approach distinguishes itself from their models in terms of how we construct the diffusion process.

# 3 Methods

Our methods have two main components. We first propose SidechainDiff (Figure 1), which is a diffusion probabilistic model for protein side-chains and can also give the structural context representations of mutations on the protein-protein interface. We then propose DiffAffinity that utilize the learned representations to predict $\Delta\Delta G$. We organize this section as follows: Section 3.1 presents the preliminaries and notations that are used throughout the paper and formally defines the problem. Section 3.2 and 3.3 describe SidechainDiff and DiffAffinity, respectively. Section 3.4 describes key details in model training.

## 3.1 Preliminaries and notations

A single-chain protein structure is composed of multiple residues (amino acids). Each amino acid shares a common central carbon atom, called the alpha ($\alpha$) carbon. The alpha carbon is bonded to an amino group (-NH2), a carboxyl group (-COOH), and a hydrogen atom, forming the amino acid's backbone. Each amino acid has a distinct atom or group bonded to the alpha carbon, known as the side-chain. A residue's side-chain structure is usually termed a side-chain conformation. The side-chain conformation varies across different amino acids, and it depends on the type of amino acid and the overall protein structure. Our main work focuses on building a generative process of side-chain conformations given the protein backbone structures.

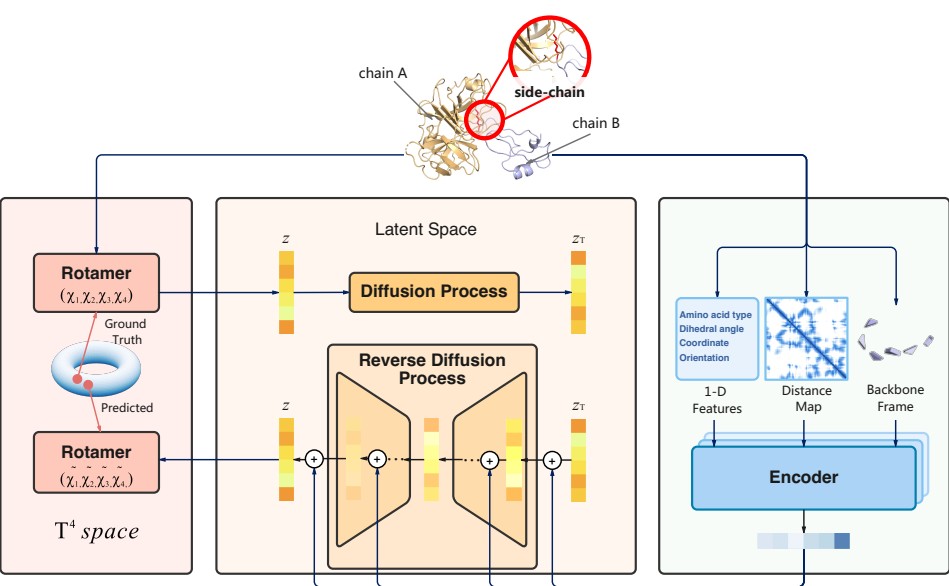

Figure 1: The overall architecture of SidechainDiff.

Multiple protein chains bind with each other and form a protein complex. Here we only consider protein complexes with two chains just for notation simplicity and note that our model works for protein complexes with more chains. For a protein complex with $n$ residues, we denote the two chains as chain $A$ and $B$, and the set of its residues as $\{1, ..., n\}$. The properities of the $i$-th amino acid include the amino acid type $a_i \in \{1, \ldots, 20\}$, the orientation $\mathbf{R}_i \in SO(3)$, the 3D coordinates $\mathbf{x}_i \in \mathbb{R}^3$, and the side-chain torsion angles $\boldsymbol{\chi}_i = (\chi_i^{(k)})_{k=1}^4$ where $\chi_i^{(k)}$ belongs to $[0, 2\pi)$.

The bond lengths and bond angles in side-chains are usually considered fixed and given backbone structures, we can solely use $\boldsymbol{\chi_i}$ to reconstruct the coordinates of each atom in a side-chain. Thus, a side-chain conformation is also called a rotamer, which is parameterized with $\boldsymbol{\chi_i}$. The number of torsional angles ranges from 0 to 4, depending on the residue type. Here, we place a rotamer in a 4-dimensional torus space $\mathbb{T}^4 = (\mathbb{S}^1)^4$ where $\mathbb{S}^1$ denotes a unit circle.

### 3.2 SidechainDiff: side-chain diffusion probabilistic model

SidechainDiff utilizes a conditional Riemannian diffusion model in $\mathbb{T}^4$ to build the generative process of the four rotamers and estimate their joint distributions (Figure 1). The generative process is conditioned on the structural context of mutations on the protein-protein interface, which is encoded using a SE(3)-invariant neural network. The learned conditional vectors serve as a natural representation of the structural context.

**Conditional diffusion probabilistic model on $\mathbb{T}^4$** We adopt the continuous score-based framework on compact Riemannian manifolds [De Bortoli et al., 2022] to construct the generative process for rotamers within $\mathbb{T}^4$. Our primary adaptation involves conditioning our diffusion model on vectors that are concurrently learned via an encoder network.

Let the random variable $\mathbf{X}$ denote the states of rotamers. And let $(\mathbf{X}_t)_{t \in [0,T]}$ and $(\mathbf{Y}_t)_{t \in [0,T]} = (\mathbf{X}_{T-t})_{t \in [0,T]}$ denote the diffusion process and associated reverse diffusion process in $\mathbb{T}^4$, respectively. The stochastic differential equation (SDE) and reverse SDE [De Bortoli et al., 2022, Song et al., 2021] can be defined as follows:

$$d\mathbf{X}_t = d\mathbf{B}_t^{\mathcal{M}}. \tag{1}$$

$$d\mathbf{Y}_t = \nabla \log p_{T-t}(\mathbf{Y}_t|\mathbf{Z})dt + d\mathbf{B}_t^{\mathcal{M}}. \tag{2}$$

Here, $\mathbf{B}_t^{\mathcal{M}}$ represents the Brownian motion on $\mathbb{T}^4$, which is approximated with a Reodesic Random Walk (GRW) on $\mathbb{T}^4$ (Algorithm 1). The score $\nabla \log p_{T-t}(\mathbf{Y}_t|\mathbf{Z})$ in Equation 2 is estimated with a

score network $s_\theta(\mathbf{X}, t, \mathbf{Z})$. We introduce the conditional vector $\mathbf{Z}$ in the score, parameterized with an encoder network that takes the structural context of mutation sites as inputs. We will delve into these components separately in the following subsections.

**Geodesic Random Walk in $\mathbb{T}^4$**   When approximating the Brownian motion on the compact Riemannian manifolds using GRWs, a crucial step is defining the projection map and the exponential map that establishes the connection between the ambient Euclidean space and the associated tangent space, as discussed in previous studies [Jørgensen, 1975, De Bortoli et al., 2022]. In the context of $\mathbb{T}^4$, which is constructed as the quadruple product of $\mathbb{S}^1$ within the ambient Euclidean space $\mathbb{R}^2$, it is adequate to focus on modeling Brownian motion on $\mathbb{S}^1$, defined as follows:

$$\mathrm{d}\mathbf{x}_t = \mathbf{K} \cdot \mathrm{d}\mathbf{B}_t, \quad \mathbf{K} = \begin{pmatrix} 0 & -1 \\ 1 & 0 \end{pmatrix}. \tag{3}$$

Here, $\mathbf{B}_t$ represents the Brownian motion on the real line in the ambient Euclidean space $\mathbb{R}^2$, and $\mathbf{K}$ denotes the diffusion coefficient.

Similarly, the projection map $\mathrm{Proj}_{\mathbf{Y}_k^\gamma}$ within the tangent space $T_{\mathbf{Y}_k}(\mathbb{T}^4)$ and the exponential map $\mathrm{Exp}$ on $\mathbb{T}^4$ can be derived by applying the Cartesian product of the maps designed for $\mathbb{S}^1$. To be more precise, these two maps are defined as follows:

$$\exp_{\boldsymbol{\mu}}(\mathbf{v}) = \cos(\|\mathbf{v}\|)\boldsymbol{\mu} + \sin(\|\mathbf{v}\|)\frac{\mathbf{v}}{\|\mathbf{v}\|}. \tag{4}$$

$$\mathrm{proj}_{\boldsymbol{\mu}}(\mathbf{z}) = \mathbf{z} - \frac{<\boldsymbol{\mu}, \mathbf{z}>}{\|\boldsymbol{\mu}\|^2}\boldsymbol{\mu}. \tag{5}$$

Here, $\boldsymbol{\mu} \in \mathbb{S}^1$, $\mathbf{v} \in T_{\boldsymbol{\mu}}(\mathbb{S}^1)$, and $\mathbf{z} \in \mathbb{R}^2$.

---

**Algorithm 1** Geodesic Random Walk (GRW) during the sampling phase

---

**Require:** $T, N, Y_0^\gamma, P, \{a_i, \mathbf{R}_i, \mathbf{x}_i, \boldsymbol{\chi}_i\}_{i=1}^{127}$
1: $\gamma = T/N$                      ▷ Step-size
2: $\mathbf{Z} = \mathrm{Enc}_{\phi^*}(\{a_i, \mathbf{R}_i, \mathbf{x}_i, \boldsymbol{\chi}_i\}_{i=1}^{127})$     ▷ Encode the structural context into hidden representation
3: **for** $k \in \{0, \dots, N-1\}$ **do**
4:    $\bar{\mathbf{R}}_{k+1} \sim \mathrm{N}(0, \mathrm{Id})$          ▷ Standard Normal noise in ambient space $(\mathbb{R}^2)^4$
5:    $\mathbf{R}_{k+1} = \mathrm{Proj}_{\mathbf{Y}_k^\gamma}(\bar{\mathbf{R}}_{k+1})$         ▷ Projection in the Tangent Space $T_{Y_k}(\mathbb{T}^4)$
6:    $\mathbf{W}_{k+1} = -\gamma s_\theta(\mathbf{Y}_k^\gamma, k\gamma, \mathbf{Z}) + \sqrt{\gamma}\mathbf{R}_{k+1}$    ▷ Compute the Euler-Maruyama step on tangent space
7:    $\mathbf{Y}_{k+1}^\gamma = \mathrm{Exp}_{\mathbf{Y}_k^\gamma}(W_{k+1})$      ▷ Move along the geodesic defined by $W_{k+1}$ and $\mathbf{Y}_k^\gamma$ on $\mathbb{T}^4$
8: **end for**
9: **return** $\{\mathbf{Y}_k^\gamma\}_{k=0}^N$

---

**Score network**   The architecture of the score network $s_\theta(\mathbf{X}, t, \mathbf{Z})$ is implemented as a three-layer Multilayer Perceptron (MLP) with 512 units in each layer. It takes as inputs the conditional vector from the encoder network representing the structural context of the mutations, the sampled rotamers, and the timestep $t$.

**Conditional encoder network**   The encoder network (Figure S1) takes as inputs both single features, consisting of amino acid type, backbone dihedral angles, and local atom coordinates for each amino acid, as well as pair features such as pair distances and relative positions between two amino acids. An SE3-invariant IPA network [Jumper et al., 2021] is utilized to obtain the conditional vector $\mathbf{Z}$ for the structural context of the mutations. Further information regarding the encoder can be found in App. A.1.

**Training objectives**   The parameters $\theta$ and $\phi$ in the score network and conditional encoder network are trained simultaneously. We adopt the implicit loss $l_t^{\mathrm{im}}(s_t)$ in De Bortoli et al. [2022], defined as follows:

$$l_t^{\mathrm{im}}(s_t) = \int_{\mathcal{M}} \{\frac{1}{2}\|s_\theta(\mathbf{X}_t, t, \mathbf{Z})\|^2 + \mathrm{div}(s_t)(\mathbf{X}_t, t, \mathbf{Z})\}\mathrm{d}\mathbb{P}_t(\mathbf{X}_t). \tag{6}$$

### 3.3 DiffAffinity: mutational effect predictor

For the mutational effect predictor, DiffAffinity, we imitate the structure of the mutated protein by setting the side-chain conformation as an empty set and altering the amino acids at the mutation

sites. Subsequently, we use the pre-trained conditional encoder $\text{Enc}_{\phi^*}$ from SidechainDiff to encode wild-type and mutant structures into distinct hidden representations. These hidden representations, following additional IPA-like transformer network and max aggregation, are fed into a Multilayer Perceptron (MLP) of only one layer for the prediction of $\Delta\Delta G$. To assess the quality of our hidden representation, we also utilize the linear combination of SidechainDiff's hidden representations to predict $\Delta\Delta G$. More model details can be found in App. A.2.

### 3.4 Model training

We train SidechainDiff with the refined experimental protein structures on the database of Protein Data Bank-REDO [Joosten et al., 2014]. To fairly benchmark the performance of SidechainDiff, we follow the same protocol in RDE [Luo et al., 2023] to preprocess the data and split the train and test sets. We train DiffAffinity with the SKEMPI2 dataset [Jankauskaitė et al., 2019]. Due to limited samples, we perform a strategy of three-fold cross-validation to train the model and benchmark the performance. Specifically, we split the SKEMPI2 dataset into three folds based on structure, ensuring that each fold contains unique protein complexes not present in the other folds. Two of the folds are utilized for training and validation, whereas the remaining fold is reserved for testing purposes. More training settings are shown in App.A.

## 4 Results

First, we assess DiffAffinity's performance in predicting $\Delta\Delta G$ using the SKEMPI2 dataset (Section 4.1) and the latest SARS-CoV-2 dataset (Section 4.2). Subsequently, we demonstrate DiffAffinity's effectiveness in optimizing antibodies, with a focus on SARS-CoV-2 as an illustrative case (Section 4.3). Additionally, we evaluate the prediction accuracy of SidechainDiff in side-chain packing.

### 4.1 Prediction of the mutational effects on binding affinity

DiffAffinity leverages the learned representation from SidechainDiff to predict mutational effects on protein-protein binding. Here, we benchmark the performance on the widely used SKEMPI2 dataset.

**Baseline models**   As mentioned in section 2.2, the models for mutational effects prediction can be categorized into energy-based, sequence-based, unsupervised, end-to-end, and pre-trained models. We have selected several representative models from each category. The selected models include FoldX [Schymkowitz et al., 2005], Rosetta [Alford et al., 2017], flex ddG [Barlow et al., 2018], ESM-1v [Meier et al., 2021], ESM-IF [Hsu et al., 2022], ESM2[Lin et al., 2023], ESM2*, DDGPred [Shan et al., 2022], and RDE-Network [Luo et al., 2023]. More details of baselines are shown in App. A.4

For ablation studies, we have also developed two variants named DiffAffinity* and DiffAffinity-Linear. DiffAffinity* employs the same model architecture as DiffAffinity but without the learned representations from SidechainDiff and it serves as an end-to-end prediction method. And DiffAffinity-Linear is a direct linear projection of SidechainDiff's hidden representations to $\Delta\Delta G$.

**Evaluation metrics**   The performance of our model is evaluated using a variety of metrics. These include traditional prediction accuracy metrics such as root mean squared error (RMSE) and mean absolute error (MAE). Given that the scalar values of experimental $\Delta\Delta G$ are not standardized, we also employ Pearson and Spearman correlation coefficients to evaluate the models' predictive capability. To assess our model's ability to identify positive or negative mutations affecting protein-protein binding, we use the AUROC and AUPRC metrics. Mutations are classified based on the sign of $\Delta\Delta G$, where positive values indicate positive impacts on binding and negative values indicate negative impacts. AUROC measures the overall discriminative power, while AUPRC accounts for imbalanced datasets.

Following the RDE-network ([Luo et al., 2023]), we also evaluate the prediction performance for each complex separately. We group mutations by complex structure, excluding groups with fewer than ten mutation data points, and derive two metrics: average per-structure Pearson correlation coefficient per-structure Pearson and average per-structure Spearman correlation coefficient per-structure Spearman.

**Experimental results**   Our methods, DiffAffinity and DiffAffinity*, outperform all baseline models across almost all metrics and achieve state-of-the-art performance on the SKEMPI2 dataset (Table 1).

Particularly for multi-point mutations, DiffAffinity notably outperforms all other methods, suggesting that our model effectively captures the crucial relationship between mutational effects and side-chain conformations, regardless of the number of mutations. DiffAffinity surpasses the performance of the RDE-Network and ESM2*, indicating that the latent representations derived from our diffusion probabilistic model, SidechainDiff, are more beneficial than those from flow-based generative models and other self-supervised pre-training representations with the same network architecture for the same task. More analysis of the three pre-training representations can be found in App C.2.

Furthermore, DiffAffinity-Linear shows that even a straightforward linear projection of latent representations can achieve performance comparable to FoldX and ESM-IF and outperform ESM-1v. The reduced performance observed with DiffAffinity*, when the SidechainDiff encoder is absent, further emphasizes the effectiveness of the latent representations extracted from SidechainDiff.

In subsequent sections (Section 4.2 and 4.3), we employ the three models trained on different splits of the SKEMPI2 dataset without any fine-tuning on downstream tasks related to predicting binding affinity. This approach allows us to assess our model's performance in diverse tasks.

Table 1: Evaluation of prediction on the SKEMPI2 dataset

| Method | Mutations | Overall | | | | | | Per-Structure | |
|---|---|---|---|---|---|---|---|---|---|
| | | Pearson | Spearman | RMSE | MAE | AUROC | AUPRC | Pearson | Spearman |
| FoldX | all | 0.319 | 0.416 | 1.959 | 1.357 | 0.671 | 0.839 | 0.376 | 0.375 |
| | single | 0.315 | 0.361 | 1.651 | 1.146 | 0.657 | 0.839 | 0.382 | 0.360 |
| | multiple | 0.256 | 0.418 | 2.608 | 1.926 | 0.704 | 0.841 | 0.333 | 0.340 |
| Rosetta | all | 0.311 | 0.346 | 1.617 | 1.131 | 0.656 | 0.810 | 0.328 | 0.298 |
| | single | 0.325 | 0.367 | 1.183 | 0.987 | 0.674 | 0.834 | 0.351 | 0.418 |
| | multiple | 0.199 | 0.230 | 2.658 | 2.024 | 0.621 | 0.798 | 0.191 | 0.083 |
| flex ddG | all | 0.402 | 0.427 | 1.587 | 1.102 | 0.675 | 0.866 | 0.414 | 0.386 |
| | single | 0.425 | 0.431 | 1.457 | 0.997 | 0.677 | 0.874 | 0.433 | 0.435 |
| | multiple | 0.398 | 0.419 | 1.765 | 1.326 | 0.669 | 0.854 | 0.401 | 0.363 |
| ESM-1v | all | 0.192 | 0.157 | 1.961 | 1.368 | 0.541 | 0.735 | 0.007 | −0.012 |
| | single | 0.191 | 0.157 | 1.723 | 1.192 | 0.549 | 0.770 | 0.042 | 0.027 |
| | multiple | 0.192 | 0.175 | 2.759 | 2.119 | 0.542 | 0.678 | −0.060 | −0.128 |
| ESM-IF | all | 0.319 | 0.281 | 1.886 | 1.286 | 0.590 | 0.768 | 0.224 | 0.202 |
| | single | 0.296 | 0.287 | 1.673 | 1.137 | 0.605 | 0.776 | 0.391 | 0.364 |
| | multiple | 0.326 | 0.335 | 2.645 | 1.956 | 0.637 | 0.754 | 0.202 | 0.149 |
| ESM2 | all | 0.133 | 0.138 | 2.048 | 1.460 | 0.547 | 0.738 | 0.044 | 0.039 |
| | single | 0.100 | 0.120 | 1.730 | 1.210 | 0.541 | 0.734 | 0.019 | 0.036 |
| | multiple | 0.170 | 0.163 | 2.658 | 2.021 | 0.566 | 0.746 | 0.010 | 0.010 |
| ESM2* | all | 0.623 | 0.498 | 1.615 | 1.179 | 0.721 | 0.887 | 0.362 | 0.316 |
| | single | 0.625 | 0.468 | 1.357 | 0.986 | 0.707 | 0.879 | 0.391 | 0.342 |
| | multiple | 0.603 | 0.529 | 2.15 | 1.67 | 0.758 | 0.909 | 0.333 | 0.304 |
| DDGPred | all | 0.630 | 0.400 | **1.313** | **0.995** | 0.696 | 0.892 | 0.356 | 0.321 |
| | single | 0.652 | 0.359 | 1.309 | 0.936 | 0.656 | 0.884 | 0.351 | 0.318 |
| | multiple | 0.591 | 0.503 | 2.181 | 1.670 | 0.759 | 0.913 | 0.373 | 0.385 |
| RDE-Net | all | 0.632 | 0.527 | 1.601 | 1.142 | 0.731 | 0.887 | 0.415 | 0.376 |
| | single | 0.637 | 0.491 | 1.341 | 0.961 | 0.720 | 0.885 | 0.413 | 0.385 |
| | multiple | 0.601 | 0.567 | 2.157 | 1.631 | 0.768 | 0.898 | 0.390 | 0.360 |
| **DA-Linear**[1] | all | 0.326 | 0.305 | 1.954 | 1.399 | 0.642 | 0.857 | 0.222 | 0.222 |
| | single | 0.318 | 0.293 | 1.649 | 1.175 | 0.651 | 0.854 | 0.209 | 0.202 |
| | multiple | 0.277 | 0.288 | 2.593 | 1.961 | 0.629 | 0.867 | 0.193 | 0.195 |
| **DiffAffinity*** | all | 0.646 | 0.538 | 1.578 | 1.113 | 0.742 | 0.741 | 0.415 | 0.392 |
| | single | 0.657 | 0.523 | 1.312 | 0.931 | **0.742** | 0.741 | 0.417 | 0.396 |
| | multiple | 0.613 | 0.542 | 2.133 | 1.606 | 0.750 | 0.750 | 0.407 | 0.379 |
| **DiffAffinity** | all | **0.669** | **0.556** | 1.535 | 1.093 | **0.744** | **0.896** | **0.422** | **0.397** |
| | single | **0.672** | **0.523** | **1.288** | **0.923** | 0.733 | **0.887** | **0.429** | **0.409** |
| | multiple | **0.650** | **0.602** | **2.051** | **1.540** | **0.784** | **0.921** | **0.414** | **0.387** |

[1] DA-Linear is short of DiffAffinity-Linear just to save space in the table.

## 4.2 Prediction of mutational effects on binding affinity of SARS-CoV-2 RBD

SARS-CoV-2 initiates infection by binding to ACE2 protein on host cells via the viral spike protein. The receptor binding domain (RBD) of the spike proteins exhibits high-affinity binding to ACE2.

The technique of deep mutational scanning has been employed to experimentally quantify the effects of all single-point mutations on the binding affinity of the ancestral Wuhan-Hu-1 RBD (PDB ID: 6M0J) [Starr et al., 2022]. These measurements have guided the survey of SARS-CoV-2 evolution and identified several significant mutation sites on RBD that substantially contribute to the binding affinity. A total of 15 significant mutation sites like NE501[3] have been identified (see App. A.5) . We predicted all 285 possible single-point mutations for these sites and calculated the Pearson correlation coefficient between the experimental and predicted $\Delta\Delta G$.

Our results show that DiffAffinity outperforms all baseline methods (Table 2), highlighting its potential to facilitate biologists in understanding the evolution of SARS-CoV-2. Furthermore, we note that DiffAffinity exhibits a substantial improvement over DiffAffinity*, indicating the effectiveness of the learned representations from SidechainDiff.

## 4.3 Optimization of human antibodies against SARS-CoV-2

The receptor-binding domain (RBD) of the SARS-CoV-2 virus spike protein (PDB ID: 7FAE) plays a pivotal role in the binding process with the host's ACE2 protein. It serves as a prime target for neutralizing antibodies against SARS-CoV-2 [Shan et al., 2022]. In our experiment, we predict all 494 possible mutations at 26 sites within the CDR region of the antibody heavy chain. We expect that predicting mutational effects on binding affinity can facilitate in identifying top favorable mutations that can enhance the neutralization efficacy of antibodies.

These mutations are ranked in ascending order according to their $\Delta\Delta G$ values, with the most favorable (lowest $\Delta\Delta G$) mutations positioned at the top. To enhance neutralization efficacy, we aim to accurately identify and rank the five most favorable mutations. DiffAffinity successfully identifies four of these mutations within the top 20% of the ranking and two within the top 10% (Table 3). DiffAffinity consistently outperforms all the baseline methods, indicating its superior ability to predict the effects of mutations. This highlights DiffAffinity's potential as a robust tool for optimizing human antibodies.

Table 2: Pearson correlation coefficient in SARS-COV-2 binding affinity

| Method | Pearson |
|---|---|
| FoldX | 0.385 |
| RDE-Net | 0.438 |
| DiffAffinity* | 0.295 |
| **DiffAffinity** | **0.466** |

Table 3: Rankings of the five favorable mutations on the human antibody against SARS-CoV-2 receptor-binding domain (RBD) by various competitive methods

| Method | TH31W | AH53F | NH57L | RH103M | LH104F |
|---|---|---|---|---|---|
| FoldX | **4.25**% | **14.57**% | **2.43**% | 27.13% | 63.77% |
| RDE-Net | **5.06**% | **12.15**% | 55.47% | 50.61% | **9.51**% |
| DiffAffinity* | **7.29**% | **0.81**% | **19.03**% | 84.21% | 28.54% |
| **DiffAffinity** | **7.28**% | **3.64**% | **18.82**% | 81.78% | **10.93**% |

## 4.4 Prediction of side-chain conformations

To assess the generative capacity of SidechainDiff straightforwardly, we employ it for predicting side-chain conformations of specific amino acids given their structural context. Our approach involves sampling rotamers from the distribution modeled by SidechainDiff and selecting the rotamer with the highest likelihood from a pool of 100 samples. For this task, we draw upon the test dataset from the PDB-REDO test split.

For a rigorous evaluation, we compare SidechainDiff with both energy-based methods, including Rosetta [Leman et al., 2020] and SCWRL4 [Krivov et al., 2009], as well as the deep learning-based methods, including RDE, AttnPacker [McPartlon and Xu, 2023], and DLPacker [Misiura et al., 2022]. All methods are run under the same settings to ensure a fair comparison. We evaluate the performance

---

[3]NE501Y denotes the substitution of asparagine (N) at position 501 on chain E with tyrosine (Y), where N and Y represent the single-letter codes for the amino acids asparagine and tyrosine, respectively.

using the Mean Absolute Error (MAE) and steric clash number of the predicted rotamers [McPartlon and Xu, 2023].

In terms of MAE for rotamer prediction, SidechainDiff outperforms energy-based methods and achieves comparable results to deep learning-based methods such as RDE and AttnPacker (Table 4 and Table S1). Notably, our method surpasses all the other methods in terms of the steric clash number. It's noteworthy that while methods like AttnPacker focus on predicting the torsion angles of static side-chain conformations, SidechainDiff emphasizes the distribution of side-chain conformations under various physical constraints. We measure the physical constraints of mutation sites using the contact number, i.e., the number of neighboring amino acids within an $8\mathring{A}$ radius. Generally, a higher contact number indicates a more constrained structural context. We observe that for mutation sites with higher constraints, SidechainDiff tends to produce more accurate side-chain conformations (Table 5). We further illustrate cases showing the trade-off between accuracy and diversity achieved by SidechainDiff under various physical constraints (Appendix D and Figure S5a-S5d).

Table 4: Evaluation of predicted side-chain conformations

| Method | $\chi^{(1)}$ | $\chi^{(2)}$ | $\chi^{(3)}$ | $\chi^{(4)}$ | Average | Clash number |
|---|---|---|---|---|---|---|
| Proportion | 100% | 82.6% | 28.2% | 12.6% | - | |
| SCWRL4 | 24.33° | 32.84° | 47.42° | 56.15° | 28.21° | 118.44 |
| Rosetta | 23.98° | 32.14° | 48.49° | 58.78° | 28.09° | 113.31 |
| RDE | **17.13°** | 28.47° | **43.54°** | 58.62° | **21.99°** | 34.41 |
| DLPacker | 21.40° | 29.27° | 50.77° | 74.64° | 26.42° | 62.45 |
| AttnPacker | 19.65° | **25.17°** | 47.61° | **55.08°** | 22.35° | 57.31 |
| **DiffAffinity** | 18.00° | 27.41° | 43.97° | 56.65° | 22.06° | **27.70** |

Table 5: Accuracy of side-chain conformations stratified based on the contact number

| Contact number | Average $\chi$ |
|---|---|
| $1 \sim 7$ | 28.70 |
| $8 \sim 10$ | 22.39 |
| $11 \sim 19$ | 15.43 |

## 5 Conclusions

We present SidechainDiff, a Riemannian diffusion-based generative model for protein side-chains. It excels in both generating side-chain conformations and facilitating representation learning. Utilizing the learned representations, our method achieves state-of-the-art performance in predicting $\Delta\Delta G$ across various test datasets, including the SKEMPI2 dataset and the latest SARS-CoV-2 dataset. Moreover, we demonstrate its effectiveness in antibody screening and its superior side-chain packing accuracy.

In future research, we can consider integrating the side-chain clash loss [Jumper et al., 2021] to refine our generative process. Additionally, we can explore building an end-to-end diffusion-based generative model for protein full-atom structures by integrating our model with existing backbone structure generation methods [Watson et al., 2023, Yim et al., 2023]. The main limitation of our model is the lack of consideration of backbone structure changes induced by mutations. Addressing the limitation could lead to enhanced performance.

## 6 Acknowledgements

We would like to thank the National Key Research and Development Program of China (2020YFA0907000), and the National Natural Science Foundation of China (32370657, 32271297, 82130055, 62072435), and the Project of Youth Innovation Promotion Association CAS to H.Z. for providing financial support for this study and publication charges. The numerical calculations in this study were supported by ICT Computer-X center.

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
