# A Implementation details

## A.1 Encoder network architecture

We adopt a similar encoder network (Figure S1) as RDE to transform the structural context of mutations in the interface to a conditional vector used by the generative process of side-chain conformations. We define the structural context as the 128 residues in closest proximity to the mutation sites. The input features can be grouped into single node features and pair edge features. The node features include amino acid types, backbone torsion angles, and local atom coordinates for each amino acid, while the edge features include pair distance and relative sequence position between two amino acids. The input features are first fed into MLP layers (denoted as Transition layer in Figure S1) and then combined with the spatial backbone frames to pass through the Invariant Point Attention Module (IPA), an SE(3)-invariant network proposed in AlphaFold2 [Jumper et al., 2021]. We use 6 IPA blocks, and the sizes of hidden representations for node features and pair features are 128 and 64, respectively.

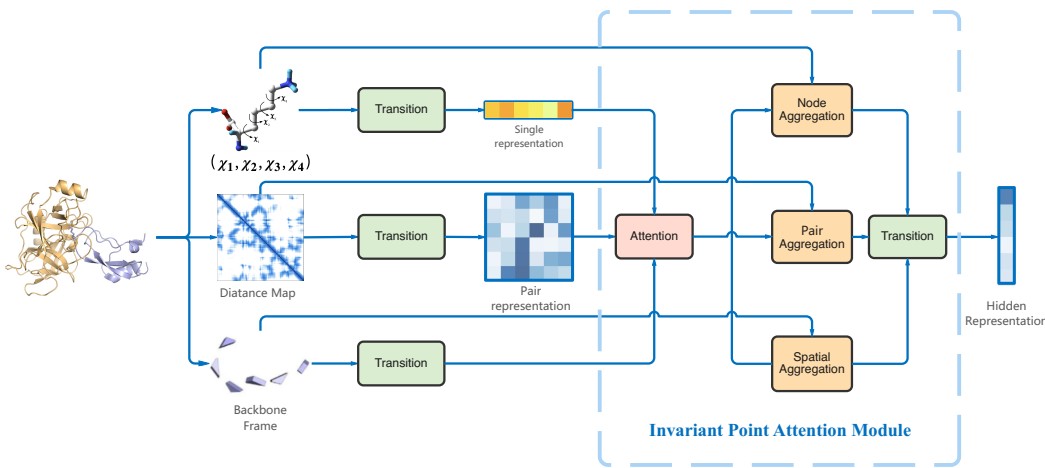

Figure S1: The architecture of the conditional encoder in SidechainDiff

## A.2 DiffAffinity network architecture

Given a wild-type $\mathcal{W}$, a mutant $\mathcal{M}$, and their structural contexts of mutations, we first obtain the hidden representations from the pre-trained SidechainDiff, denoted as $h_{\text{wt}}$ and $h_{\text{mt}}$, respectively. We set wild-type and mutant sequences as single features concated with hidden representations from SidechainDiff, distance matrix from wild-type protein structure as pair features, and frames of wild-type protein structure. Then, we input them into IPA transformer encoder [Jumper et al., 2021] to update these hidden representations. A max-pooling layer and an MLP layer follow to predict the final $\Delta\Delta G$. We use the mean squared error (MSE) loss in training.

## A.3 Training details for SidechainDiff

For SidechainDiff, we adopt a similar hyperparameter setting of the score-based generative model as used in [Song et al., 2021, De Bortoli et al., 2022]. The diffusion coefficient is parameterized as $g(t) = \sqrt{\beta(t)}$ with $\beta : t \mapsto \beta_{\min} + (\beta_{\max} - \beta_{\min}) \cdot t$, where $\beta_{\min} = 0.001$ and $\beta_{\max} = 2$. To parameterize the vector field on $\mathbb{T}^4$, we use a single field per dimension pointing in a consistent direction around the $i^{th}$ component in the product, with the unit norm. Sinusoidal activation functions are utilized.

For the ISM losses $l_t^{im}$, we adopt the setting of $\lambda_t = g(t)^2 = \beta_t$. All models are trained using the stochastic optimizer Adam with the setting of $\beta_1 = 0.9$ and $\beta_2 = 0.999$ and a batch size of $M$. The learning rate is annealed with a linear ramp from 0 to 1000 and thereafter with a cosine schedule. The total number of iterations, denoted as $N_{\text{iter}}$, is set to 200,000, and we define the batch size $M$ as

32. The reverse diffusion model is configured with 100 steps. The Algorithm 2 illustrates the entire training algorithm.

To sample mutations in training, we mask the rotamers of 10% of the amino acids and introduce noises to the rotamers of amino acids within a $C_\beta - C_\beta$ distance of $8.0\mathring{A}$ from the closest masked amino acids, simulating the impact of mutations on adjacent amino acids. Following the strategy in RDE [Luo et al., 2023], we add Gaussian noises centered at 0 and its standard deviation depending on the $C_\beta - C_\beta$ distances.

---

**Algorithm 2** Training Process of SidechainDiff

---

**Require:** $\epsilon, T, N, \{\mathbf{X}_0^m\}_{m=1}^M, \theta_0, \phi_0, N_{\text{iter}}, P, \{a_i, \mathbf{R}_i, \mathbf{x}_i, \boldsymbol{\chi}_i\}_{i=1}^{127}$
 1: ///TRAINING///
 2: **for** $n \in \{0, \ldots, N_{\text{iter}} - 1\}$ **do**
 3:     $\mathbf{X}_0 \sim (1/M) \sum_{m=1}^M \delta_{X_0^m}$             $\triangleright$ Random mini-batch from dataset
 4:     $t \sim U([\epsilon, T])$             $\triangleright$ Uniform samping between $\epsilon$ and $T$
 5:     $\mathbf{Z}_t = \text{Enc}_{\phi_n}(\{a_i, \mathbf{R}_i, \mathbf{x}_i, \boldsymbol{\chi}_i\}_{i=1}^{127})$             $\triangleright$ Encode the structural context into hidden representation
 6:     $\gamma = t/N$
 7:     **for** $k \in \{0, \ldots, N\}$ **do**             $\triangleright$ Approximate forward diffusion with Algorithm 1
 8:         $W_{k+1} \sim \mathcal{N}(0, \text{Id})$
 9:         $W_{k+1} = \sqrt{\gamma} W_{k+1}$             $\triangleright$ the drift coefficient of Algorithm 1 is set as 0 here
10:         $X_{k+1} = \exp_{\mathbf{X}_0}(W_{k+1})$
11:     **end for**
12:     $\mathbf{X}_t = X_N$
13:     $l_t^{\text{im}}(s_t) = l_t^{\text{im}}(s_{\theta_n}(\mathbf{X}_t, t, \mathbf{Z}_t))$             $\triangleright$ Compute implicit score matching loss
14:     $\theta_{n+1}, \phi_{n+1} = \text{optimizer\_update}(\theta_n, \phi_n, l_t^{\text{im}}(s_t))$             $\triangleright$ ADAM optimizer step
15: **end for**
16: $\theta^* = \theta_{N_{\text{iter}}}, \phi^* = \phi_{N_{\text{iter}}}$
17: **return** $\theta^*, \phi^*$

---

## A.4 Baseline models

To benchmark the performance, we train RDE and the two variants of our DiffAffinity (i.e. DiffAffinity* and DiffAffinity-Linear) using the same splits of training and test set with the SKEMPI2 dataset. The implementation details of baseline methods are described below.

**DiffAffinity-Linear**   DiffAffinity-Linear model represents a simple linear projection of the learned representations from SidechainDiff for the prediction of $\Delta\Delta G$.

**DiffAffinity\***   In contrast to the original DiffAffinity, no learned representations from SidechainDiff are used in DiffAffinity*. Other settings including model architecture and training procedure are the same with DiffAffinity.

**RDE** [Luo et al., 2023]   We use the training and testing script in the RDE GitHub repository (https://github.com/luost26/RDE-PPI). And for downstream tasks, we average the predictions from 3 models as the final scores.

**Rosetta** [Alford et al., 2017]   We use Rosetta version 2023.35 downloaded from the official site (https://www.rosettacommons.org).For a mutated structure, we build its structure using the CARTESIAN_DDG command. $\Delta\Delta G$ is determined by subtracting the energy of the wild-type from that of the mutant predicted by INTERFACE_ENERGY.

**FoldX** [Schymkowitz et al., 2005]   We use FoldX-v5 downloaded from the official site (https://foldxsuite.crg.eu/).For a mutated sequence, we build its structure using the BUILDMODEL command. $\Delta\Delta G$ is determined by subtracting the energy of the wild-type from that of the mutant.

**flex ddG** [Barlow et al., 2018]   We employed the flex ddG from the GitHub repository found at https://github.com/Kortemme-Lab/flex_ddG_tutorial. The binding affinity was derived using the DEFAULT setting as outlined in the tutorial.

**ESM-1v** [Meier et al., 2021]   We use the testing script of ESM-1v in the ESM GitHub repository (https://github.com/facebookresearch/esm). We derive the scores using MASKED-MARGINAL mode to serve as the metric for $\Delta\Delta G$.

**ESM-IF** [Hsu et al., 2022]  We failed in running the ESM-IF for the very large complex structures. And in Table 1, we just use results obtained from the published work which benchmarks the performance in the same testing set [Luo et al., 2023].

**ESM2** [Lin et al., 2023]  We employ a test script available in the ESM GitHub repository (`https://github.com/facebookresearch/esm`). The model takes mutant and wild-type sequences as input and produces hidden representations of these sequences using ESM2. The difference between these hidden representations is then passed through a two-layer MLP (Multilayer Perceptron) to predict the change in free energy ($\Delta\Delta G$). Here, we have employed the ESM2 (3B) language model, with the network parameters of the MLP set at $2560 \times 64 \times 1$. The training was conducted using the Adam optimizer with a learning rate of 5e-4, completing 30,000 training steps.

**ESM2\*** [Lin et al., 2023]  We also utilize the hidden representations from ESM2, similar to the ESM2 model. However, in this case, the hidden representations from the SidechainDiff model are replaced with the hidden representations from ESM2. These modified hidden representations, along with the ESM2 hidden representations, are fed into the DiffAffinity model, which predicts the change in free energy ($\Delta\Delta G$).

**DDGPred** [Shan et al., 2022]  It's very challenging to reproduce the training and testing process of DDGPred. First, no training scripts are provided in the DDGPred GitHub repository (`https://github.com/HeliXonProtein/binding-ddg-predictor`). Second, the model weights provided in the open-source repository are trained on a set that overlaps with the testing set in our work. Thus in Table 1, we just use results obtained from the published work which benchmarks the performance in the same testing set [Luo et al., 2023] and we have not benchmarked the performance of DDGPred in the downstream tasks.

## A.5 Dataset of SARS-CoV-2 RBD binding affinity

In the previous study [Starr et al., 2022], 15 crucial mutational sites have been identified that greatly influence SARS-CoV-2 RBD binding affinity. The sites include NE501, SE477, GE339, NE440, TE478, SE373, QE498, EE484, SE371, QE493, GE496, YE505, GE446, SE375, and KE417. We use all 285 possible single-point mutations on these sites to benchmark the performance.

# B  Source code

Code and data are available at `https://github.com/EureKaZhu/DiffAffinity/`

# C  Additional results of DiffAffinity on the SKEMPI2 dataset

## C.1  Performance of DiffAffinity on the SKEMPI2 dataset under different Per-Structure threshold

To analyze the robustness and accuracy of our performance across different per-structure thresholds, we show the Pearson and Spearman correlation under various per-structure thresholds from 1 to 20 (Figure S2a-S2b). DiffAffinity achieves state-of-the-art results compared with other methods under all thresholds.

## C.2  Analysis of different pre-training representations

To demonstrate the efficacy of the representations learned by SidechainDiff, we performed PCA (Principal Component Analysis) to reduce the dimensions of the obtained representations from SidechainDiff on the SKEMPI2 dataset and visualized the distribution of the representations (Figure S3a).

Furthermore, we have compared several representative methods, including RDE's representations based on flow models (Figure S3b) and ESM2's representations based on protein language models (Figure S3c). It can be observed that the representations obtained by SidechainDiff are capable of more effectively distinguishing data under different $\Delta\Delta G$ values. Although ESM2 exhibits outstanding performance in tasks such as protein secondary structure prediction and protein contact

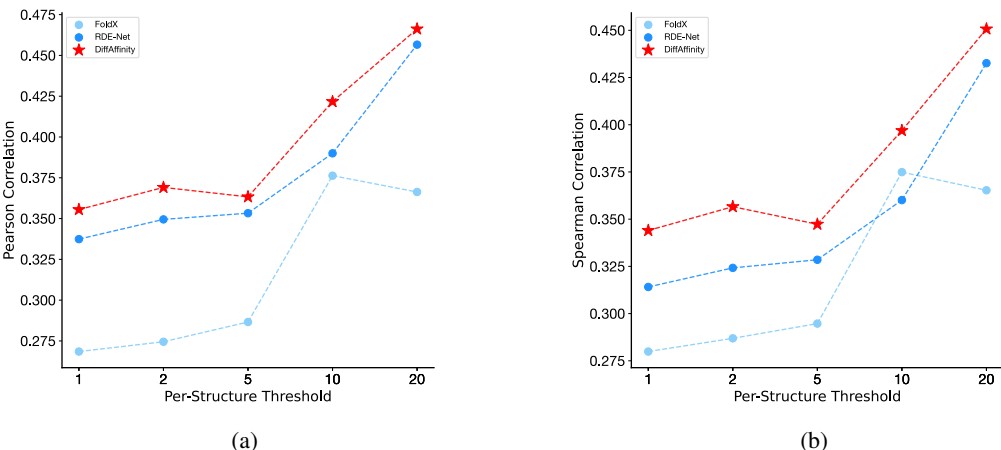

(a)                                           (b)

Figure S2: The performance of Pearson and Spearman correlations across different per-structure filtering thresholds.

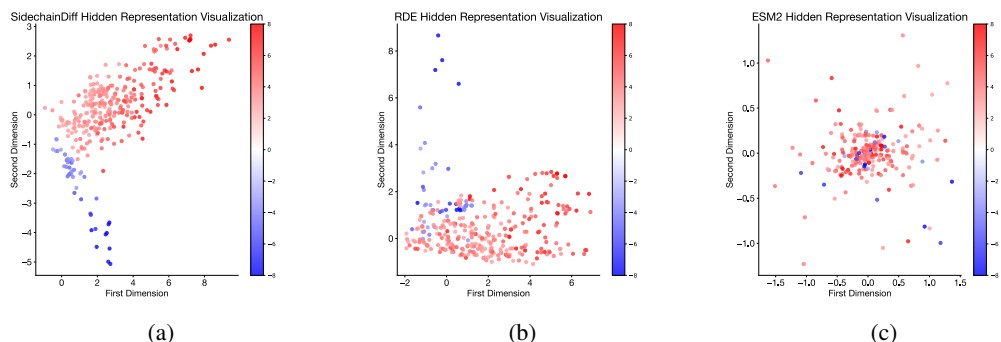

(a)                          (b)                          (c)

Figure S3: Visulization of pre-training hidden representations on the SKEMPI2 dataset. We utilize the different colors to represent different $\Delta\Delta G$ obtained by experiment.

recognition, it is insensitive in predicting the effects of mutations on the binding affinity of protein complexes.

### C.3 Figures on the performance with the SKEMPI2 dataset

To visually demonstrate the predictive capability of DiffAffinity on the SKEMPI2 dataset, we employed scatter plots and histograms to showcase the statistical properties of results derived from our model DiffAffinity. It is evident that DiffAffinity precisely captures the statistical distribution across the entire SKEMPI2 dataset, including both its single-mutation and multi-mutations subsets (Figure S4).

## D  Additional results of SidechainDiff

### D.1  Prediction side-chain conformation error with types of amino acids

We show the error of side-chain conformations' prediction for each amino acid side-chain conformations in the test dataset of PDB-REDO in Table S1. In the majority of amino acids as shown, SidechainDiff surpasses energy-based methods (SCWRL4, Rosseta) and exhibits performance comparable to that of RDE.

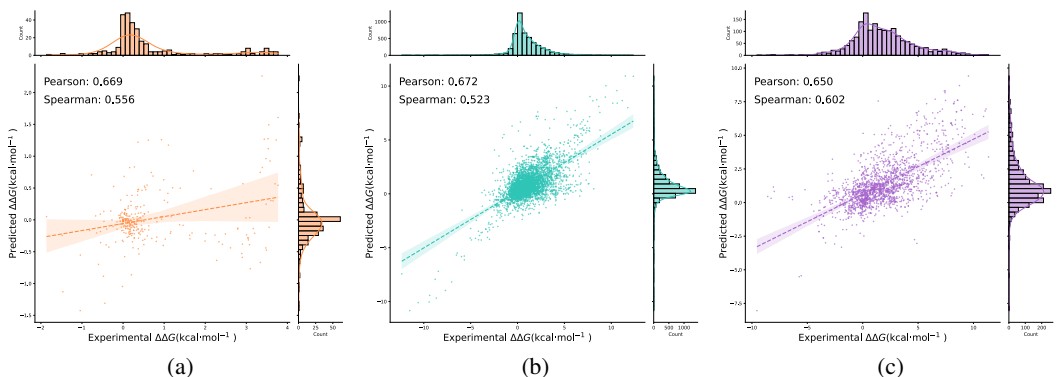

Figure S4: Results on SKEMPI2 dataset. (a) Correlation between experimental $\Delta\Delta G$s and DiffAffinity predictions on the whole SKEMPI2 dataset. (b) Correlation between experimental $\Delta\Delta G$s and DiffAffinity predictions on the single-mutation subset of SKEMPI2 dataset. (c) Correlation between experimental $\Delta\Delta G$s and DiffAffinity predictions on the multi-mutations subset of SKEMPI2 dataset.

Table S1: Mean absolute error of the predicted side-chain torsion angle

| Type | $\chi$ | SCWRL4 | Rosetta | RDE | SidechainDiff | Type | $\chi$ | SCWRL4 | Rosetta | RDE | SidechainDiff |
|------|--------|--------|---------|-----|---------------|------|--------|--------|---------|-----|---------------|
| ARG | 1 | 29.30 | 30.50 | 18.92 | **18.60** | LEU | 1 | 13.97 | 14.15 | 10.83 | **8.46** |
| | 2 | 28.68 | 36.12 | **27.36** | 33.55 | | 2 | 23.76 | 27.61 | 24.39 | **21.14** |
| | 3 | 57.89 | 60.73 | **51.40** | 51.73 | LYS | 1 | 31.32 | 33.88 | 30.73 | **24.55** |
| | 4 | **60.35** | 63.62 | 62.76 | 61.82 | | 2 | **30.95** | 33.15 | 36.77 | 36.78 |
| ASN | 1 | 21.70 | 19.39 | **11.30** | 17.10 | | 3 | 38.90 | 42.07 | 41.71 | **37.20** |
| | 2 | 44.00 | 43.41 | 39.25 | **37.24** | | 4 | 51.94 | 53.94 | 54.89 | **51.56** |
| ASP | 1 | 25.75 | 22.62 | 18.05 | **17.24** | MET | 1 | 26.36 | 26.07 | **16.95** | 25.33 |
| | 2 | 23.90 | 21.26 | **20.35** | 20.95 | | 2 | 38.52 | 36.09 | **29.24** | 30.15 |
| CYS | 1 | 24.83 | 25.90 | 16.75 | **7.33** | | 3 | 55.11 | 58.77 | **46.33** | 58.43 |
| GLN | 1 | 33.16 | 31.53 | 26.37 | **26.32** | PHE | 1 | 12.30 | 13.08 | **9.64** | 12.39 |
| | 2 | 46.33 | **33.96** | 45.48 | 37.88 | | 2 | 12.40 | 12.35 | **10.02** | 11.14 |
| | 3 | 53.72 | 56.52 | **52.26** | 60.59 | SER | 1 | 47.19 | 46.83 | 35.48 | **27.51** |
| GLU | 1 | 35.45 | 34.69 | **21.56** | 35.56 | THR | 1 | 28.05 | 22.67 | **13.26** | 21.09 |
| | 2 | 38.23 | 38.43 | **36.53** | 41.75 | TRP | 1 | **14.52** | 18.64 | 14.69 | 17.29 |
| | 3 | 31.50 | **30.85** | 34.47 | **28.23** | | 2 | 31.74 | 31.44 | 31.39 | **26.01** |
| HIS | 1 | 23.15 | **19.12** | 29.12 | 25.69 | TYR | 1 | 11.39 | 14.56 | **6.83** | 10.33 |
| | 2 | 70.62 | 61.97 | 74.55 | **54.74** | | 2 | 11.37 | 14.45 | **7.96** | 8.34 |
| ILE | 1 | 13.92 | 14.65 | 10.45 | **7.84** | VAL | 1 | 21.31 | 19.41 | **12.89** | 15.69 |
| | 2 | 26.43 | 27.54 | 26.84 | **23.29** | | | | | | |

## D.2 Diversity of sampled side-chain conformations

The diversity and prediction accuracy of sampled side-chain conformations from SidechainDiff highly correlates with the structural constraints presented in the protein-protein interface (Table S2). We quantify the diversity using the entropy of the sampled side-chain conformations while utilizing the contact number as a surrogate for the extent of structural constraints. The contact number of an amino acid is defined as the count of neighboring residues within a $C_\beta - C_\beta$ distance of $8\mathring{A}$. Amino acids with higher contact numbers tend to exhibit greater structural constraints, indicating a more constrained conformation. In highly constrained regions, the sampled side-chain conformations exhibit lower entropy and higher prediction accuracy. These observations align consistently with previous studies in the field [Jones and Thornton, 1996].

Table S2: The diversity of side-chain conformations with different structural constraints

| Contact number | Average contact numbers | Average error of $\chi$ | Entropy |
|:---:|:---:|:---:|:---:|
| $1 \sim 7$ | 5.57 | 28.70 | 4.10 |
| $8 \sim 10$ | 8.86 | 22.39 | 3.76 |
| $11 \sim 19$ | 12.36 | 15.43 | 3.33 |

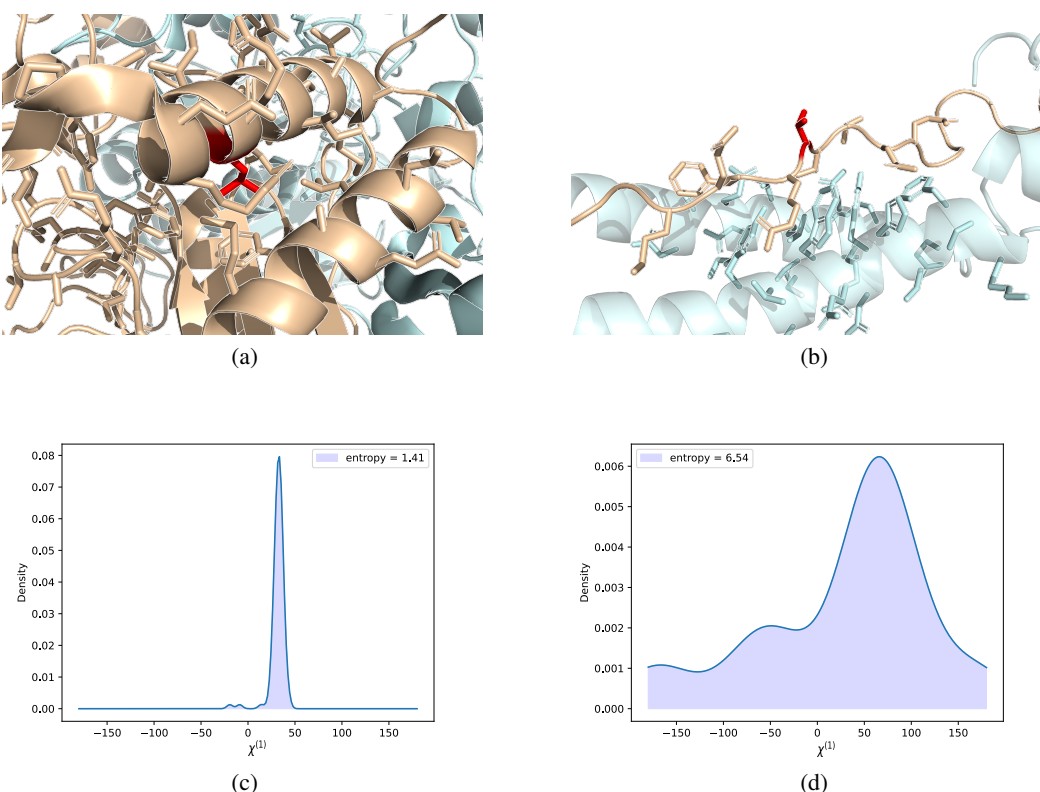

Figure S5: (a) The structural context of L302 on the chain A of the protein complex 6HBV. (b) The structural context of T28 on the chain D of the protein complex 2P22. (c) The distribution of $\chi^{(1)}$ for the sampled side-chain conformations of L302 on the chain A of the protein complex 6HBV. (d) The distribution of $\chi^{(1)}$ for the sampled side-chain conformations of T28 on the chain D of the protein complex 2P22.

We further present two illustrative cases that highlight the distinction between highly constrained and less constrained regions (Figure S5). The first case involves L302 on the chain A of the protein complex 6HBV (Figure S5a). In this instance, the structural context is characterized by a high degree of constraint, resulting in sampled side-chain conformations with an entropy of 1.67 (Figure S5c). In

contrast, the second case focuses on T28, located in the loop region of chain D of the complex 2P22 (Figure S5b). In this scenario, the sampled side-chain conformations display much more flexibility, as indicated by an entropy value of 6.54 (Figure S5d).

Here, we then specify the details that how to calculate the entropy and average error of $\chi$ in Table S2. The entropy $S$ is defined by the Boltzmann expression:

$$S = -k_B \int p(\mathbf{x}) \log p(\mathbf{x}) \mathrm{d}\mathbf{x} = -k_B \mathbb{E}_{\mathbf{x} \sim p} \log p(\mathbf{x}),$$

where $p(\mathbf{x})$ refers to the distribution of conformation $\mathbf{x}$ and $k_B$ denotes the Boltzmann constant. We assign a value of 1 to $k_B$ just for simplicity. The entropy $S$ is then calculated as the mean of the log probabilities over 100 samplings.

We calculate the weighted average error of all side-chain torsion angles as follows:

$$\text{Average error of torsion angles} = \frac{1}{N} \sum_{i=1}^{N} \frac{\sum_{j=1}^{4} p(\chi^{(j)}) \mathbb{I}_i(\chi^{(j)}) e_i(\chi^{(j)})}{\sum_{j=1}^{4} p(\chi^{(j)}) \mathbb{I}_i(\chi^{(j)})},$$

where

$$p(\chi^{(j)}) = \frac{1}{N} \sum_{i=1}^{N} \mathbb{I}_i(\chi^{(j)}), \quad j = 1, 2, 3, 4;$$

$$\mathbb{I}_i(\chi^{(j)}) = \begin{cases} 1 & \text{if } \chi^{(j)} \text{ exists in the } i\text{th amino acid} \\ 0 & \text{otherwise} \end{cases}.$$

Here, $e_i(\chi^{(1)})$, $e_i(\chi^{(2)})$, $e_i(\chi^{(3)})$, and $e_i(\chi^{(4)})$ respectively denote the errors of the predicted $\chi^{(1)}$ to $\chi^{(4)}$ in the $i$-th sampling, while $N$ represents the total number of amino acids.