# OpenReview forum: "Predicting mutational effects on protein-protein binding via a side-chain diffusion probabilistic model"
_NeurIPS.cc/2023/Conference — NeurIPS 2023 poster_

### Official Review · Reviewer_HbdY · 2023-07-03

**Soundness:** 3 good
**Presentation:** 2 fair
**Contribution:** 3 good
**Rating:** 7
**Confidence:** 4

**Summary:**

The paper proposes SidechainDiff a diffusion model over the torsion angles of the protein sidechains and uses the learned representations of this model to predict the mutational effects of protein-protein binding.

**Strengths:**

The paper shows a wide variety of promising experimental results ranging from standard benchmarks of binding affinity and side-chain conformation to case studies related to SARS-CoV-2 RBD and antibodies. I cannot comment on their reproducibility because the code was not provided.


**Weaknesses:**

While the experimental results are strong, however for the paper to be ready for publication, the manuscript should be improved in many regards: comparison to the literature, presentation of the method and certain baseline comparisons.

Comparison to the literature:

1. The paper lacks references to existing published methods that have developed Riemannian diffusion models over the hyperdimentional torus to model molecular torsion angles (e.g. [1] and [2]). In these regards, the authors should clarify the relationship between the diffusion process presented here and in those works. To my understanding, the diffusion processes are identical which raises the question of why the authors refer to the perturbation kernel as intractable and resort to the use of implicit score matching. [1] suggests that the perturbation kernel can be sampled analytically (with a truncated infinite series). Are the perturbation kernels the same? If so how does the explicit score matching approach of [1] compare to the one developed?
2. Related to the point before how are the samples X_t and the scores s_t obtained in equation (7)?
3. Except from the diffusion component, the overall approach of the paper seems analogous to RDE [3]. I believe that given the similarity it would be useful to clarify the differences between the methods. For example line 256 claims the two methods use the same architecture. Is this true and if so does doing the pretraining with the flow of RDE-Network get worse performance than no pretraining at all (DiffAffinity*)?

Presentation of the method:

4. For the network architecture hardly any details are provided in the main text and even in the appendix these are not very clear. The authors should add some details of the architecture to the main text and clarify the presentation in the appendix: e.g. how is the score of the chi angles predicted? Is it an MLP on top of the hidden representation?

Baseline comparisons:

5. The authors should add details about the hyperparameter and model tuning process. E.g. were any hyperparameters tuned for the retrospective studies on SARS-Cov-2?
6. Given the high-dimensional distribution with complex interdepencies between the angles of the different sidechains, looking simply at the MAE of individual torsion angles seems inadequate. This is especially relevant when some angles have reported MAE above 40 degrees, e.g. what would be the MAE performance of a simple baseline of predicting the median (on the circle) for every residue type and angle? More generally, I believe the authors should also provide further metrics, for example what is the steric clash rate in each of the approches?
7. Some claims in the paper seem not well justified and should be adjusted. E.g. line 305 (and similar 322) “[SidechainDiff] achieves comparable performance to RDE, suggesting superior capability in generating side-chain conformations”, if they are comparable then the method is not superior.

Minor:

8. The text presents many orthographic errors (e.g. to name just a few at lines 161, 224, 549) I suggest the authors to use publicly available programs to review the text and correct these.

[1] Jing, Bowen, et al. "Torsional diffusion for molecular conformer generation." *NeurIPS 2022*.

[2] Corso, Gabriele, et al. "Diffdock: Diffusion steps, twists, and turns for molecular docking." ICLR 2023.

[3] Luo, Shitong, et al. "Rotamer Density Estimator is an Unsupervised Learner of the Effect of Mutations on Protein-Protein Interaction." ICLR 2023.

**Questions:**

See weaknesses section.

**Limitations:**

See weaknesses section.

---

> ### Author Rebuttal · Authors · 2023-08-07
>
> We sincerely appreciate the reviewer's constructive feedback. We have carefully addressed each comment and made the necessary revisions accordingly.
>
> __Q1:__
> 1. Thanks for your advice, We have incorporated references to existing published methods [1, 2] in the related work section.
> 2. Comprehensive analysis between our method and their methods [1, 2] can be found in **Q6 in the "global" response**.
>
> 3. Due to time constraints, we were unable to perform a comprehensive comparison between the explicit score matching approach of [1, 2] and the implicit score matching approach used in our paper. In our setting, ISM loss doesn't need to calculate the ground truth score function in torus space $\mathbb{T}^{4}$ which is consistent with our method, so we choose ism loss as our loss function. We acknowledge that the ism loss has its disadvantages, particularly in its reliance on the calculation of score network divergence, which can be challenging.
>
> __Q2:__
>  We use a random walk sample procedure to get $X_t$ which means pertubated $X_0$ . The forward diffusion procedure can be expressed as : $dX_t = d\mathbf{B}_t$. To be specialized, the forward diffusion procedure involves the following steps:
>
>  1. Sample a point $r$ from a standard normal distribution in $(\mathbb{R}^2)^4$.
>  2. Use the exponential map to project $r$ onto the tangent space of $X_0$.
>  3. Scale the tangent vector by a factor of $\sqrt{t}$.
>  4. Project the scaled tangent vector back to the unit circle using the projection map.
>  5. Obtain the point $X_t$ on the unit circle.
>
>  We can formalize this sampling procedure on the 4-dimensional torus space $\mathbb{T}^4$ using the following equation:
>
>  $X_t = Proj_{X_0}(\sqrt{t} Exp_{X_0}(r)), r\sim \mathcal{N}(0,I_{(\mathbb{R}^2)^4}).$
>
> In Equation 7, We apologize that $s_t$ is a typo. we mean $s_\theta$ acturally.  $s_\theta$ is score network.
>
> __Q3:__
> 1. We thank the reviewer for pointing out the error in that model architecture. We adopted the same architecture to the RDE-Network for the $\Delta\Delta G$ stage. However, it is important to note that we employed a completely different model to handle the side-chain conformation. We use the same architecture to predict binding affinity but use different hidden representations from pre-training modules. Our performance is better than RDE-Network, which indicates that our pre-training module is more suitable for mutation-related tasks.
>
> 2. Yes, DiffAffinity* has better performance than RDE-Network in SKEMPI2 dataset. In our experiments, we observed that DiffAffinity* without any pre-training features outperformed RDE-Network. Unfortunately, the original paper [2] does not provide the results of RDE-Network without RDE hidden representation, preventing us from directly verifying whether our results align with the original findings. Nevertheless, in downstream tasks of SARS-CoV-2 RBD and human antibodies, we observed a significant improvement in the performance of RDE-Network over DiffAffinity*. This suggests that the pre-training features from RDE could indeed enhance the model's generalization ability and the end-to-end model DiffAffinity* might overfit the SKEMPI2 dataset.
>
>  A comprehensive analysis of our hidden representation obtained from SidechainDiff and the comparation with that from RDE and ESM2 can be found **in Q4 in the "global" response and in Figure a in the PDF**.
>
> __Q4:__
> The architecture of the score network $s_{\theta}(\mathbf{X}_t,t,\mathbf{Z})$ is implemented using a multi-layer perceptron (MLP) with 3 layers, each containing 512 units. In DiffAffinity, we concat the hidden representation from SidechainDiff and sequence information as the sequential input of the IPA-like transformer.
> We have incorporated additional details regarding our network architecture in both the main text and the appendix.
>
> __Q5:__
> We appreciate the point raised by the reviewer on the issue of model hyperparameters. We do not employ any fine-tuning techniques in SARS-CoV-2 RBD mutational effects prediction and optimization of human antibodies against SARS-CoV-2 tasks. Instead, we utilize the models trained on the SKEMPI2 dataset and directly use their predictions for the $\Delta\Delta G$ of mutations in downstream tasks.
>
> __Q6:__
> 1. We have incorporated more metircs and methods [3, 4] of side-chain conformation **in Q6 in the "global" response and in Table c in the PDF**.
> 2. Additionally, we experimented to predict the median (on the circle) for each residue type and angle, and the results are presented in the table below.
>
>     | Method | $\chi_1$ | $\chi_2$ | $\chi_3$ | $\chi_4$ |
>     |--------|---------|---------|---------|---------|
>     | Random | 89 | 98 | 68 | 114.64 |
>
> __Q7:__
> We have made the necessary corrections to rectify all the writing mistakes in our paper. We sincerely appreciate your valuable advice and guidance
>
> __Reference__
>
> [1] Jing, Bowen, et al. "Torsional diffusion for molecular conformer generation." Advances in Neural Information Processing Systems 35 (2022): 24240-24253.
>
> [2] Corso, Gabriele, et al. "Diffdock: Diffusion steps, twists, and turns for molecular docking." ICLR 2023.
>
> [3] Luo, Shitong, et al. "Rotamer Density Estimator is an Unsupervised Learner of the Effect of Mutations on Protein-Protein Interaction." bioRxiv (2023): 2023-02.
>
> [4] McPartlon, Matthew, and Jinbo Xu. "An end-to-end deep learning method for protein side-chain packing and inverse folding." Proceedings of the National Academy of Sciences 120.23 (2023): e2216438120.
>
> [5] Misiura, Mikita, et al. "DLPacker: deep learning for prediction of amino acid side chain conformations in proteins." Proteins: Structure, Function, and Bioinformatics 90.6 (2022): 1278-1290.

---

> > ### Comment · Reviewer_HbdY · 2023-08-15
> > **Response to rebuttal**
> >
> > Thank you for the careful response. I believe that the additions of these theoretical analyses and better presentation of the method will be very valuable additions to the paper which retains promising experimental results. I have therefore raised my score to 7 and recommend acceptance.

---

> > > ### Author Response · Authors · 2023-08-19
> > >
> > > We thank the reviewer again for the constructive comments on our work.

---

### Official Review · Reviewer_Z2H5 · 2023-07-05

**Soundness:** 4 excellent
**Presentation:** 3 good
**Contribution:** 4 excellent
**Rating:** 8
**Confidence:** 3

**Summary:**

The paper introduces a new representation learning-based approach called SidechainDiff that predicts how amino acid mutations influence protein-protein binding using Riemannian diffusion model. It's very important problem in the field of structural biology, for example, in evaluation of antibody variants impact to antigen binding. Moreover, SidechainDiff is the first approach that focuses on side chains compared to previous methods that work with protein backbones. Overall, this paper is a valuable contribution to both computational biology and artificial intelligence.

**Strengths:**

Originality: The paper provides a new solution for a well-known task of the prediction of amino acid mutation impact into binding. Intoduced methods are new (SidechainDiff and DiffAffinity), though authors used an already known concept of diffusion models. Current work focuses on side chain generation while previous studies provides protein backbone generation. The related studies are adequately cited.
Quality: The submission is technically sound, appropriate, and complete. It provides a clear and detailed explanation of the all steps of the work. In addition, the authors compare DiffAffinity with other models providing a baseline of its performance. The authors provide insights into the further steps of the work and potential applications.
Clarity: The submission is written clearly and organized. It provides all the necessary details for the reader.
Significance: Due to the lack of experimentally determined structures there is a need for tools that are able to predict the effect of amino acid mutations to the binding without this information. This paper provides such a tool.

**Weaknesses:**

The predicted mutational effect was checked only on one type of antibody-antigen system (SARS-CoV-2 RBD). It's not clear if the model will show the same performance  in other antibody-antigen systems or how the performance will change. Also, it's not discuss how the protein characteristics (such as length, mutation position, etc.) can influence the performance and predictions.

**Questions:**

1. Will SidechainDiff be an open source approach? If so, it would be very useful to prepare a GitHub repository with the code and instruction of how to use the code.
2. Can the results of SidechainDiff be biased towards SARS-CoV-2 RBD? Have you checked the results on other antibody-antigen systems? Will results remain the same?

**Limitations:**

The limitations of SidechainDiff are not clearly written, please provide detailed discussion

---

> ### Author Rebuttal · Authors · 2023-08-06
>
> We sincerely appreciate the reviewer's constructive comments on our work. In response, we have carefully addressed their concerns and provided clarification on various issues as follows:
>
> __Q1:__
> To obey the rules of the double-blind reviewing policy, we initially don't provide a GitHub URL. But SidechainDiff will be released soon after the peer review finishes, and we will provide the training, prediction, and instruction of SidechainDiff.
>
> __Q2:__
> DiffAffinity is also practical in other antigen and antibody systems. In response to the question, we introduce additional tasks in other antibody-antigen systems. We aim to predict deep-sequences libraries of the therapeutic antibody trastuzumab for specificity to human epidermal growth factor receptor 2 (HER2) with experimentally validated binary labels. For this purpose, we utilize DiffAffinity, RDE, and ESM2* to discriminate between antigen-binding or non-binding mutations in the CDR H3 region of the 1N8Z antibody protein, where the total number of mutations is 491 [1]. The mutations are classified based on the sign of $\Delta \Delta G$, where positive values indicate antigen-binding and negative values indicate non-binding interactions. To address the imbalanced dataset, we employ the AUPRC metric. The results of this task are presented in the table below.
> | Method | AUPRC |
> |----|----|
> |ESM2*|0.739|
> |RDE|0.755|
> |DiffAffinity| 0.768|
>
> Compared with other deep learning methods, our model DiffAffinity also achieves the best performance in this task which indicates our model can easily handle the different datasets.
>
> __Q3:__
> We acknowledge that one of the disadvantages of our model is its inability to capture the changes in backbone structure caused by mutations.
>
> __Reference__
>
> [1] Mason, et al. Optimization of therapeutic antibodies by predicting antigen specificity from antibody sequence via deep learning. Nat Biomed Eng (2021)

---

> > ### Comment · Reviewer_Z2H5 · 2023-08-17
> >
> > Thank you for the response! I'm satisfied with the additions.

---

> > > ### Author Response · Authors · 2023-08-19
> > >
> > > We thank the reviewer again for the constructive comments on our work.

---

### Official Review · Reviewer_U583 · 2023-07-07

**Soundness:** 3 good
**Presentation:** 2 fair
**Contribution:** 2 fair
**Rating:** 5
**Confidence:** 4

**Summary:**

The paper introduces DiffAffinity, a novel method for predicting the effects of mutations on protein-protein interactions. The authors leverage a diffusion model to learn representations that aid in predicting changes in binding affinity (∆∆G). They compare DiffAffinity's performance with existing methods such as FoldX and RDE-Net. The paper also explores the application of DiffAffinity in optimizing human antibodies against SARS-CoV-2.

**Strengths:**

1. The paper innovatively proposes a diffusion-based approach for predicting sidechain conformations at the protein-protein interface.
The authors provide a thorough comparison of DiffAffinity with existing methods, demonstrating its superior performance.
2. The authors provide a thorough comparison of DiffAffinity with existing methods, demonstrating its superior performance.

**Weaknesses:**

1. The paper's main claim—that the hidden representation learned from the diffusion model can enhance binding affinity prediction—is not clearly elaborated.
2. The proposed method does not account for changes in the backbone, which could limit its effectiveness in cases where mutations cause such changes.
3. The paper could benefit from a discussion on the computational efficiency of DiffAffinity compared to other methods.

**Questions:**

1. Could the authors elucidate why the representation learned from the diffusion-based model benefits binding affinity prediction? More elaboration on this point would be beneficial.
2. Is there a performance comparison between the model using diffusion-based representation and the model using other self-supervised representations (e.g., ESM2)? It would be insightful to see the model performance when replacing the diffusion-based representation with other representations.
3. The score network receives timestep t as input. When computing the hidden representation for the downstream task of affinity prediction, what's the value of t, and why was it chosen?
4. After obtaining the hidden representation from the diffusion-based encoder, an additional module predicts the binding affinity from the hidden representation. Is this additional module also an IPA-like transformer? The appendix seems to lack the specific model architecture of this module.
5. In Table 1, the model appears to have achieved state-of-the-art performance without the representation learned from the diffusion model. Does this suggest that the performance gain primarily stems from the model architecture itself?
6. Could the authors elaborate on the computational efficiency of DiffAffinity? How does it scale with the size of the protein or the number of mutations?

**Limitations:**

The proposed method only considers side chain conformation changes and does not take backbone changes into account. This limitation might hinder its practical applications.

---

> ### Author Rebuttal · Authors · 2023-08-07
>
> We appreciate the reviewer's approval and constructive comments on our work. In response, we have carefully addressed their concerns and provided clarification on various issues as follows:
>
> __Q1:__
> The affinity of the complex is derived from the dynamic conformation of the protein rather than its precise fixed structure. We employ a diffusion model to characterize the distribution of side-chain conformations, which allows for a more accurate simulation of the true physical interactions of the side chains.
>
> To elucidate the benefits of the representations learned by the diffusion model for affinity prediction, we incorporated an experiment visualizing these representations after dimensionality reduction using PCA (**Q4 in the "global" response and Figure a in the PDF**). In comparison with the representations from the flow model (RDE) and the pretrained language model (ESM-3B), we found that the representations based on the Sidechain Diff can more directly discern the effects of mutations on affinity.
>
> __Q2:__
> We conducted additional comparisons between our DiffAffinity model and two other baselines: (a) ESM2 with a 2-layer MLP predictor, and (b) feeding ESM2 representations to the DiffAffinity architecture. Importantly, our DiffAffinity model consistently outperformed these baselines in all datasets mentioned.
> The results of these comparisons can be found in **Table a in the PDF**. The findings from these comparisons align with the conclusions drawn from addressing Q1, both indicating that the hidden representation obtained from SidechainDiff is more suitable for mutation-related tasks. These results further validate the efficacy and superiority of our DiffAffinity model in handling mutation-related challenges.
>
> __Q3:__
> We thank the reviewer for pointing out the potential confusion caused by the usage of the score network, in the DiffAffinity model, we exclusively use the conditional encoder from SidechainDiff, where the inputs are wild-type sequences and wild-type protein structures to obtain hidden representations. These representations are then fed into DiffAffinity. Consequently, we do not utilize the score network from SidechainDiff, and there is no need to consider the timestep $t$ for the score network in this context.
>
> In the sampling procedure, we employ the score network to generate samples from the reverse process. The specifics of the sampling procedure and the setting of parameter $t$ can be found in **Algorithm 1 of Section 3.2**.
>
> __Q4:__
> Yes, the additional module is an IPA-like transformer. The DiffAffinity model utilizes the hidden representation from SidechainDiff, along with sequence and protein structure information, as input to an IPA-like transformer. This transformer generates the final representation, and the difference between the final representations for the wild-type and mutation is used to predict the $\Delta \Delta G$ value. We have now added a detailed description in **Section A.2 of Appendix**.
>
> __Q5:__
> In our experiments, we observed that DiffAffinity* without any pre-training features outperformed RDE-Network and other methods. Unfortunately, the original paper [1] does not provide the results of RDE-Network without RDE hidden representation, preventing us from directly verifying whether our results align with the original findings. Nevertheless, in downstream tasks of SARS-CoV-2 RBD and human antibodies, we observed a significant improvement in the performance of DiffAffinity over DiffAffinity*. This suggests that the pre-training features from SidechainDiff can indeed enhance the model's generalization ability and the End-to-End model DiffAffinity* may overfit the SKEMPI2 dataset. Meanwhile, we keep the same hyperparameter and model architecture with RDE-Network and ESM2* in the second stage of DiffAffinity. Our model shows superior performance than RDE-Network and ESM2* in downstream tasks which also indicates our hidden representation is more suitable for mutation-related tasks.We extend our analysis about pre-training features in **Q4 in the "global" response**.
>
> __Q6:__
> Thanks for your attention to computational efficiency, according to our preprocess procedure, we first crop all protein complexes into patches containing 128 residues by first choosing a seed residue, and then choose 127 nearest neighbors based on C-beta distances. Therefore, the computational efficiency of DiffAffinity is not affected by the size of the protein or the number of mutations from the model design. We also experiment to verify this conclusion.
> | Number of mutations | Time |
> |----|----|
> | 1 | 2.33s|
> | 2~5 | 2.33s|
> | >6 | 2.33s|
>
> __Q7:__
> We have now acknowledged the limitations of our model in the **Conclusion section** in the revision.
>
>
> __Reference__
>
> [1] Luo, Shitong, et al. "Rotamer Density Estimator is an Unsupervised Learner of the Effect of Mutations on Protein-Protein Interaction." bioRxiv (2023): 2023-02.

---

> > ### Comment · Reviewer_U583 · 2023-08-18
> >
> > Thank you for addressing my comments and queries in your rebuttal. I appreciate the effort you've invested in responding to my concerns. However, I still have some concern regarding the newly added ESM2 representation and the DiffAffinity experiment. Specifically, it seems counterintuitive that the ESM2 representation combined with the DiffAffinity model performs worse than the DiffAffinity model alone, without a pre-trained representation. Could you provide an explanation for this surprising result?
> >
> > In addition, while the SARS-CoV-2 experiment shows that DeepAffinity has a significantly better Pearson correlation coefficient than DeepAffinity*, the rankings for the five favorable mutations seem very close. How should we interpret this apparent discrepancy in the experimental results?

---

> > > ### Author Response · Authors · 2023-08-19
> > >
> > > We appreciate the reviewer's approval and constructive comments on our work. In response, we have carefully addressed the remaining concerns as follows:
> > >
> > >
> > > __Q1__:
> > >
> > > 1.The inefficacy of ESM embedding for this task is mainly due to the fact that the ESM language model is trained only on the sequences of protein single chains rather than the protein complexes.
> > >
> > > 2.The dimension of ESM2 embedding is 2,560, considerably larger than the 128 dimensions of the representation embeddings used in our DiffAffinity and the previous method RDE. The scarcity of curated training labels for this task might lead to overfitting with higher-dimensional representations. It could explain that ESM2 representation combined with the DiffAffinity model performs worse than the DiffAffinity model alone.
> > >
> > > 3.Additionally, to assess the utility of ESM2 embeddings, we applied PCA to visualize these embeddings. Our observations revealed their failure to effectively capture the magnitudes of $\Delta\Delta G$ (more details in the global response Q2 and figure a in the global PDF).
> > >
> > > Our findings are consistent with previous studies, such as RDE [1] (Table 1 in its original paper), which also indicated the poor performance of ESM language model in predicting mutational effects on protein-protein binding.
> > >
> > > __Q2__:
> > >
> > > We also understand the concern of the reviewer about the SARS-CoV-2 experiment results.
> > >
> > > We first clarify that **Table 2** and **Table 3** show evaluations on two distinct tasks. Specifically, **Table 2** presents the evaluation on the **mutations within the RBD region** of SARS-CoV-2’s spike protein.  This dataset comprises hundreds of mutations, and we employed the Pearson correlation coefficient to assess the accuracy of the predicted DDG.
> > > On the other hand, **Table 3** focuses on the evaluation on **mutations within the antibodies** that bind to the spike protein for SARS-CoV-2. Due to the absence of experimentally determined $\Delta\Delta G$ results in this dataset, we followed previous works to evaluate the methods’ performance  in identifying and ranking the top 5 favorable mutation sites for binding.
> > >
> > > The results in **Table 3** indicate that DiffAffinity outperforms DiffAffinity*. However, due to very limited amount labeled data available for this task, substantial differences between these methods may not be evident.
> > >
> > > [1] Luo et al. Rotamer Density Estimator is an Unsupervised Learner of the Effect of Mutations on Protein-Protein Interaction. ICLR (2023)

---

### Official Review · Reviewer_wzHr · 2023-07-25

**Soundness:** 2 fair
**Presentation:** 1 poor
**Contribution:** 2 fair
**Rating:** 5
**Confidence:** 4

**Summary:**

The paper studies the problem of mutational effect prediction on protein-protein binding. To address the data scarcity issue, the authors propose to first pre-train their model on PDB using diffusion models to fit the protein side-chain distribution. Then, the learned representations are used for predicting the mutational effects on SKEMPIv2. The method achieves good results on the benchmark in the author’s evaluation setting.

**Strengths:**

1. Mutational effects on protein-protein binding is an important and difficult problem in the protein community. It suffers from the data scarcity issue and pre-training seems like a good solution to the problem.
2. The paper introduces diffusion-based models to fit the protein side-chain distribution, which, to the best of my knowledge, is among the first papers to explore this approach.

**Weaknesses:**

1. The paper's writing and clarity are poor, making it hard to follow. Many sections seem to be directly copied from previous works. The authors introduce many unnecessary notations and equations, which makes the method even more difficult to understand.
2. The novelty of the paper is limited, as it primarily extends existing works without introducing problem-specific designs. The substitution of flow-based models with diffusion models for side-chain distribution modeling is not sufficiently innovative.
3. The related work section lacks a discussion of existing methods on protein side-chain packing, which would provide a more comprehensive context for readers.
4. The experiment design contains significant flaws that undermine the persuasiveness and support of the evaluation presented. Notably, many important baselines and details are missing from the experiment section, which weakens the arguments made by the authors. (See questions)

**Questions:**

Major points:
1. A significant portion of Section 3.2 appears to be directly copied from the Riemannian diffusion model paper [1], including Propositions 1 and 2. Please discuss why the two propositions are important to side-chain problems and mention in the main paper that **these contributions are derived from previous works**. Also, the section should be rewritten with appropriate paraphrasing to clarify the changes made to adapt the model for the new task.
2. The authors propose to use a three-fold cross-validation for evaluation, but several details regarding the dataset split need to be addressed. (a) How is the dataset split? Is it split by proteins or mutations? Is it split based on sequence similarity or structure similarity? (b) The claim that DDGPred does not provide training scripts, and the authors copied the results from the published work is not a valid excuse. The training script of DDGPred is very easy to implement. Reproducing the results with the same dataset splits is essential to ensure fair comparison, as the performance can vary significantly with different splits due to limited data. Cross-validation should be repeated multiple times to assess the significance of the improvement.
3. Important baselines are missing in the benchmark. First, more traditional methods should be included in comparison, e.g. FlexDDG [2]. These methods hardly suffer from the overfitting problem. Second, the performance of using pre-trained representations with a linear predictor is low and the improvement of DiffAffinity over DiffAffinity* is small. This raises the doubt on the effects of pre-training. Please include a baseline using other pre-trained representations, e.g., ESM2. Two baselines should be considered (a) ESM-2 with a 2-layer MLP predictor and (b) Feeding ESM-2 representations to the DiffAffinity architecture.
4. The reported metrics are problematic. First, AUROC is more suitable for balanced binary classification tasks, and the number of positive and negative samples should be reported for binary classification tasks. If the data is imbalanced, AUPRC should also be reported. Second, treating mutations with ddG around 0 as either positive or negative is not appropriate, considering the measurement errors and the fact that most mutations in SKEMPIv2 are neutral. Additionally, discarding proteins with fewer than 10 mutations when reporting per-structure Pearson raises concerns about the sensitivity of the reported metric to this threshold. The authors should clarify the number of proteins and mutations left for evaluation and provide justifications for choosing 10 as the threshold.
5. The evaluation in Sec. 4.3 is not convincing. Assigning high ranks to only five experimentally tested mutations is insufficient to draw conclusions about the effectiveness of the proposed method, especially considering the limited number of mutations tested experimentally in the original paper [3]. The authors should consider a more comprehensive evaluation by testing a larger set of possible mutations.
6. In Sec. 4.4, the authors only compare their method with three baselines and ignore recent advancements in side-chain packing problem [4, 5]. Since modeling side-chain conformation is a major contribution of the paper, more experimental evaluation with a broader set of baselines should be included to demonstrate the method's effectiveness.

Minor points:
1. Line 121: Please give a definition of backbone atoms and side-chain atoms. This will be more friendly to audience without knowledge about proteins.
2. Eq. (3) in Sec. 3.2: $U(X_t)$ is used without any definition.
3. Line 148: Please discuss how the Brownian motion on $\mathrm{T}^4$ is defined in the context of side-chain packing problem.
4. Eq. (4) in Sec. 3.2: The SDE here is Variance-Exploding SDE, while the concepts introduced before are based on Variance-Preserving SDE. Please correct me if I am wrong.
5. Line 165: Please discuss why the perturbation kernel $p_{t|0}$ is difficult to obtain.
6. Line 171: “equation 6 -> Equation 6” Please make the references to equations consistent across the paper.

Overall, I think the paper is not ready to publish in its current state. However, I firmly believe that the paper holds great potential for making a valuable contribution to the field if the authors dedicate more efforts towards improving the experimental design and writing. By addressing the weaknesses and incorporating the suggested changes, the paper can reach a higher standard of quality and become a valuable addition to the existing literature. With the necessary revisions, I am confident that the paper can be transformed into a strong and impactful publication.

[1] De Bortoli, Valentin, et al. "Riemannian score-based generative modelling." Advances in Neural Information Processing Systems 35 (2022): 2406-2422.

[2] Barlow, Kyle A., et al. "Flex ddG: Rosetta ensemble-based estimation of changes in protein–protein binding affinity upon mutation." The Journal of Physical Chemistry B 122.21 (2018): 5389-5399.

[3] Shan, Sisi, et al. "Deep learning guided optimization of human antibody against SARS-CoV-2 variants with broad neutralization." Proceedings of the National Academy of Sciences 119.11 (2022): e2122954119.

[4] Misiura, Mikita, et al. "DLPacker: deep learning for prediction of amino acid side chain conformations in proteins." Proteins: Structure, Function, and Bioinformatics 90.6 (2022): 1278-1290.

[5] McPartlon, Matthew, and Jinbo Xu. "An end-to-end deep learning method for protein side-chain packing and inverse folding." Proceedings of the National Academy of Sciences 120.23 (2023): e2216438120.

**Limitations:**

The authors should add a separate paragraph to dicuss the potential limitations.

---

> ### Author Rebuttal · Authors · 2023-08-06
>
> We thank the reviewer for constructive comments on our work. Below are our responses to the reviewer's concerns.
>
> __Q1:__
> We realize that too many notations similar to the paper of the Riemannian Diffusion Model may be misleading. We agree with the reviewer in this aspect. Following the reviewer’s suggestion, we have rewritten Section 3.2.
>
> In the initial submission, we included these two propositions and explicitly cited their work just for the completeness of our paper. To avoid misleading, we now have put these propositions in the supplementary and clearly state their reference.
>
> To clarify the change for our model, the primary modifications of the original Riemannian diffusion model for our task are presented in **Q2 in the "global" response**.
>
> __Q2:__
> 1. We split the dataset by proteins based on structure similarity. Specifically, we have split the SKEMPI2 dataset into three folds based on structure, ensuring that each fold contains unique protein complexes not present in the other folds. Two of the folds are utilized for training and validation, whereas the remaining fold is reserved for testing purposes.
> 2. Following the reviewer’s suggestion, we have retrained the DDGPred model using the same hyperparameters as outlined in the reference. The result indicates that the performance of DDGPred is roughly comparable with the reported in their reference (please see **Table a in the PDF**).
> 3. Following the reviewer's suggestion, we repeated the training of our model 5 times in cross-validation experiments. The results demonstrate that our methods are robust and stable in the test (Please see the **Table b in the PDF**).
>
> __Q3:__
> 1. In the previous submission, we selected the most representative and publicly available benchmarks from different method categories. As the reviewer pointed out, we have checked FlexDDG according to their reference. But FlexDDG was not accessible, and no server was available for evaluation
> 2. Following the reviewer's suggestion, we performed comparisons between DiffAffinity and two baselines. The new results can be found in **Table a in the PDF**. For an in-depth analysis of SidechainDiff, we extend to a visualization analysis of the baseline results using PCA (Please see **Q4 in the "global" response and in Figure a in the PDF**).
>
> __Q4:__
> 1. Following the reviewer's suggestion, we report the new AUPRC metrics behind. The AUPRC results demonstrate that our model outperforms all the baselines, indicating its effectiveness in handling imbalanced datasets.
>     | Method | FoldX | ESM-1v | ESM-IF| DDGPred| RDE-Net | Linear| DiffAffinity*| DiffAffinity|
>     |-|-|-|-|-|-|-|-|-|
>     |AUPRC|0.839|0.735|0.768|0.892|0.887|0.741|0.857|0.896|
> 2. Following the reviewer's suggestion, we have reported the results considering only mutations with $\Delta \Delta G$ numeric values above 1 or below -1. Our model exhibits consistent performance with previous results (please see **Table e in the PDF**).
> 3. Following the RDE setting, we choose 10 as per-structure thresholds. We clearly show that our model consistently outperforms all the baselines, regardless of the chosen threshold ( please see **Table d in the PDF**).
>
> __Q5:__
> 1. In addition to the task in Sec. 4.3, we also evaluated our method on another downstream task, including 285 mutations (as described in Sec. 4.2).  Our method achieves a Pearson correlation of 0.466, higher than other methods.
> 2. Following the reviewer’s suggestion, we further evaluate our method on the HER2-antibody data including 491 mutations with experimentally validated binary labels[4]. Our methods also achieve better performance than RDE.
> | Method | AUPRC |
> |----|----|
> |ESM2*	|0.739|
> |RDE|0.755|
> |DiffAffinity| 0.768|
>
> __Q6:__
> Following the reviewer's suggestion, we have included AttnPaker and DLPaker for comparison. The analysis of the updated results can be found in **Q5 in the "global" response and Table c in the PDF**.   Our method achieves better performance than AttnPaker and DLPacker in terms of both average MSE of side-chain torsion angles and side chain clash number.
>
> __Q7: Minor points__
> 1. We have now added a definition of backbone atoms and side-chain atoms in the previous submission.
> 2. Thank you for the guidance. We have removed the irrelevant notation $U(X_t)$ from Eq. (3) as it does not pertain to our tasks.
> 3. In the context of the side-chain packing problem, we represent the rotamer (side-chain conformation) in $\mathbb{T}^4 = (\mathbb{S}^1)^4$. Consequently, we initially consider the Brownian motion in $\mathbb{S}^1$ and then extend it directly to $\mathbb{T}^4$.The diffusion equation for the density of Brownian particles $\mathbf{Y}$ at point $x$ and time $t$ on the unit circle is given by:
> $\mathrm{d}\mathbf{Y} = -\frac{1}{2}\mathbf{Y}\cdot\mathrm{d}t + \mathbf{K}\cdot\mathbf{Y}\cdot\mathrm{d}\mathbf{B}_t$
>
>    $\quad K_{11}=K_{22}=0,K_{12}=-1,K_{21}=1$. Here, $\mathbf{B}_t$ represents the Brownian motion on the real line. By applying the diffusion process to each dimension and taking the Cartesian product of the results, we obtain the Brownian Motion in the 4-dimensional  torus space $\mathbb{T}^4$.
>
> 4. Yes, the SDE in our paper is Variance-Exploding SDE.
>
> 5. We have now provided more details about the perturbation kernel in **Q6 in the "global" response**.
>
> 6. We have corrected the writing errors in our paper.
>
> __References__
>
> [1] Shan et al. Deep learning guided optimization of human antibody against SARS-CoV-2 variants with broad neutralization. Proceeding of the National Academy of Sciences (2022)
>
> [2] Luo et al. Rotamer Density Estimator is an Unsupervised Learner of the Effect of Mutations on Protein-Protein Interaction. ICLR (2023)
>
> [3] Starr et al. Shifting mutational constraint in the SARS-CoV-2 receptor-binding domain during viral evolution. Science (2022)
>
> [4] Masonet et al. Optimization of therapeutic antibodies by predicting antigen specificity from antibody sequence via deep learning. Nat Biomed Eng (2021)

---

> > ### Comment · Reviewer_wzHr · 2023-08-16
> >
> > I'd like to thank the authors for their detailed response. However, my concerns about the experimental setting remain. Here are my response:
> >
> > >Q2
> > 1. According to your description, the dataset is split by protein complexes, but it is not split by structure similarity. "Split by structure similarity" means the structures in training set and test set have structure similarity below a pre-defined threshold. If no sequence or structure similarity constraints are added during dataset split, the task will become much easier.
> > 2. Thanks for adding this baseline. The results look reasonable.
> > 3. The variance reported in Table b is too small. I guess the authors just retrain the model on the same split with different random seeds. However, "repeat cross-validation multiple times" means that the dataset split should be regenerated every time and the model should be retrained on the new splits. According to my experience, this will yield very large variance in the model performance and even change the rank of different methods. This avoids that the proposed method only excels on a certain split, which is very common on small datasets.
> >
> > >Q3
> > 1. FlexDDG is a very strong baseline on this task, which should not be neglected. It is publicly available as a Rosetta script, the tutorial of which can be found in https://github.com/Kortemme-Lab/flex_ddG_tutorial.
> > 2. Thanks for adding experiments with ESM representations. It is quite surpurising to see that DiffAffinity with ESM2 performs even worse than DiffAffinity with random initialization. Also, my question "the performance of using pre-trained representations with a linear predictor is low and the improvement of DiffAffinity over DiffAffinity* is small. This raises the doubt on the effects of pre-training" is still not addressed.
> >
> > >Q4
> > 1. The improvement of DiffAffinity over DDGPred in terms of AUPRC is too small.
> >
> > Overall, I'd like to thank the authors' promise to revise the paper and parts of my concerns about the experimental setting are addressed. However, the major concerns still remain: (1) lack of repeated experiments (2) lack of important baselines (3) the incremental improvement brought by pre-training. Therefore, I keep my score unchanged.

---

> > > ### Author Response · Authors · 2023-08-20
> > >
> > > We appreciate the reviewer’s constructive comments on our work. Below are our responses to the reviewer's remaining concerns.
> > >
> > > __Q2__
> > > 1. We follow the strategy of splitting the train-test dataset used in prior works such as RDE [1] and GeoPPI [2]. To elucidate the structural similarity in SKEMPI2 dataset split, we run TM-align on each pair of protein complexes between the training and testing folds. Of these, 98.6% have a TM-score below 0.3 and only 0.6% have a TM-score exceeding 0.6. It is worth noting that a TM-score below 0.4 indicates a very low level of structural similarity [3].
> > >
> > >    Furthermore, we would like to emphasize that,  beyond the evaluation on SKEMPI2 dataset through  cross-validation, we assessed the performance on independent testing sets, including the set of mutations on RBD of SARS-CoV-2’s spike protein (**Table 2**) and the set of mutations on the antibodies of the SARS-CoV-2 (**Table 3**).  Our method demonstrates state-of-the-art results on these independent testing sets.
> > >
> > > 3. We acknowledge that we misunderstood the reviewer’s suggestion in the previous revision, where we simply retrained the model on the same split with different random seeds. Following the reviewer’s suggestions, we have now retrained our model 5 times with different data splits. We show details on standard deviation and mean values below. DiffAffinity consistently outperformed RDE in terms of average scores across all evaluation metrics.
> > >     | Method | Pearson $\mu$ | Pearson $\sigma$  | Spearman $\mu$ | Spearman $\sigma$ | AUPRC $\mu$  | AUPRC $\sigma$ |
> > >     |-|-|-|-|-|-|-|
> > >     |RDE|0.616| 0.032 |0.508|0.025|0.883|0.002|
> > >     |DiffAffinity|0.668|0.027|0.547|0.026|0.893|0.001|
> > >
> > > __Q3__:
> > > 1. Following the review’s suggestion, we have now included FlexDDG for comparison, as presented below. DiffAffinity outperforms FlexDDG on all metrics.
> > >    | Method | Pearson | Spearman |AUROC| AUPRC |Per-Structure Pearson|Per-Structure Spearman
> > >    |-|-|-|-|-|-|-|
> > >    |FlexDDG|0.402|0.427|0.675|0.866|0.414|0.386
> > >
> > > 2. i)The inefficacy of ESM embedding for this task is mainly due to the fact that the ESM language model is trained only on the sequences of protein single chains rather than the protein complexes.
> > >
> > >    ii)The dimension of ESM2 embedding is 2,560, considerably larger than the 128 dimensions of the representation embeddings used in our DiffAffinity and the previous method RDE. The scarcity of curated training labels for this task might lead to overfitting with higher-dimensional representations. It could explain that ESM2 representation combined with the DiffAffinity model performs worse than the DiffAffinity model alone.
> > >
> > >    Our findings are consistent with previous studies, such as RDE [1] (**Table 1** in its original paper), which also indicated the poor performance of ESM language model in predicting mutational effects on protein-protein binding.
> > >
> > >    We understand the reviewer's concerns about the relative improvements from the pre-training model.
> > >
> > >    i)  It's worth highlighting that DiffAffinity outperforms DiffAffinity* across all evaluation metrics, as detailed in Table 2. Notably, it exceeds DiffAffinity* by 4.6% on the AUPRC metric (**Table in Q4**).
> > >
> > >    ii).Compared to DiffAffinity*, DiffAffinity demonstrates substantial improvement when evaluated on the independent testing set for SARS-CoV-2 (**Table 2**). Specifically,  DiffAffinity  and DiffAffinity * achieve  Pearson correlation coefficients of 0.466 and 0.295, respectively. The performance on an independent testing set is more indicative of the model's  generalization capability.
> > >
> > > __Q4__:
> > >
> > > 1. We conducted a comprehensive evaluation of the methods utilizing a range of metrics widely-used in previous work. DiffAffinity achieves overall better performance than DDGPred. While DiffAffinity did not demonstrate a substantial improvement over DDGPred in terms of the AUPRC metric, it exceeds DDGPred by 18.6% in terms of Spearman correlation coefficients. Furthermore, for both the Per-Structure Pearson correlation coefficient sand the Per-Structure Spearman correlation coefficients, DiffAffinity exhibits notable increases of 12.5% and 11.1%, respectively (refer to **Table 1**).
> > >
> > >    Additionally, it's important to note that DiffAffinity is much more computationally efficient than DDGPred.
> > >
> > >
> > > [1] Luo et al. Rotamer Density Estimator is an Unsupervised Learner of the Effect of Mutations on Protein-Protein Interaction. ICLR (2023).
> > >
> > > [2] Liu et al. Deep geometric representations for modeling effects of mutations on protein-protein binding affinity. Plos Computational Biology (2021).
> > >
> > > [3] Xu et al.  How significant is a protein structure similarity with TM-score=0.5?. Bioinformatics (2010).

---

> > > > ### Comment · Reviewer_wzHr · 2023-08-20
> > > >
> > > > I'd like to thank the authors' diligent work during rebuttal. The response now addresses most of my concerns.
> > > >
> > > > However, I'm still not convinced by the experiments on SARS-CoV-2, where only limited data are used for evaluation. Please note that **following the setting in previous paper does not mean the experimental setting is convincing**. To show the potential of your method on real-world applications, you need the wet-lab experiments, not just ranking mutants selected by other methods. Also, given the fact that DDGPred is able to identify the good mutations for better binding affinity, the contribution of this experiment is diminished.
> > > >
> > > > Considering the novelty and presentation problem in the paper, I decide to raise my score from 3 to 5.

---

> > > > > ### Author Response · Authors · 2023-08-21
> > > > >
> > > > > We sincerely appreciate the reviewer's valuable feedback on our work and are grateful for the raised score.
> > > > >
> > > > > We understand the reviewer's concerns regarding the antibody screening experiment for SARS-CoV-2 [1]. In this experiment, we utilized previously validated variants from wet-lab experiments as the ground truth to evaluate the performance of our method. While we acknowledge the importance of validating additional top-ranked variants identified by our approach through wet-lab experiments, we wish to clarify that this aspect falls beyond the scope of the research presented in this paper. Our primary focus is on the development of the computational methodology. Furthermore, it's important to note that our other results highlight the broader application of DiffAffinity in various aspects of protein-protein interactions beyond antibody screening.
> > > > >
> > > > > [1] Shan, Sisi, et al. "Deep learning guided optimization of human antibody against SARS-CoV-2 variants with broad neutralization." Proceedings of the National Academy of Sciences 119.11 (2022): e2122954119.

---

### Author Rebuttal · Authors · 2023-08-06

In our “global” response, to address reviewers’ main concerns,  we elucidate our motivation, delineate the distinctions between our approach and previous methods (including the original Riemannian diffusion model and torsion diffusion model), and present new results (including comparison with ESM2, visualization analysis of the learned representation, and expanded method comparisons for side-chain conformations)

__Q1: Motivation__

Given limited annotated experimental data, we utilize representation learning on unlabeled data to improve mutational effect prediction in protein-protein binding. Recognizing the inherent flexibility of side-chain conformations, we introduce a conditional diffusion model to capture the dynamic nature of side-chain conformations, rather than relying on a fixed conformation. Our downstream task results and visualization analysis (**Figure a in the PDF**) of learned representations demonstrate its effectiveness.

__Q2: Difference with Riemannian Diffusion Model__

Our primary modifications to adapt the original Riemannian diffusion model for our task are as follows:
1. We extend the original Riemannian diffusion model to a conditional diffusion model, where we jointly learn the conditional vector and the diffusion process. Specifically, we encode the structural context of the mutation as a conditional vector and learn the generative process of its side-chain conformations in a 4-dimensional torus space.

2. We explore the model's capacity for representation learning. Side-chain conformations are inherently flexible, with the degree of flexibility dependent on their structural context. Therefore, we hypothesize that a representation learning approach, aimed at learning the distribution and generative process of side-chains rather than a point estimation, can lead to improved performance in downstream tasks.


__Q3: Additional Benchmark of ESM2__

We included pre-trained language model baselines in SKEMPI2 dataset. We conducted comparative experiments with ESM2-3B + 2-layer MLP and ESM2-3B + DiffAffinity (**Table a in the PDF**). Notably, DiffAffinity achieved state-of-the-art results on almost all benchmarks within the SKEMPI2 and SARS-CoV-2 datasets.

__Q4: Pre-training representation Analysis__

To assess the pre-trained representative capacity of SidechainDiff based on the diffusion model (**Figure a (a) in the PDF**), we compared it with other methods RDE (**Figure a (b) in the PDF**) and ESM2-3B (**Figure a (c) in the PDF**).
We calculated the difference of hidden representations between wild-type and mutant proteins obtained from pre-training methods and applied PCA to reduce the dimensions of the representations from SKEMPI2. We visualized the distribution of the representations and colored them based on their $\Delta\Delta G$ values.
Compared to other pre-trained methods, the representations from our model can more effectively discern affinity changes caused by mutations in the latent space.


__Q5: Side-chain Conformation Results__

We provided a sidechain conformation comparison with baselines based on side-chain packing methods, such as AttnPacker [1] and DLPacker [2]. Our model achieves comparable results with these methods which aim to predict side-chain conformations accurately.
The results can be found in **Table c in the PDF**. We also discuss the diversity of side-chain conformation generated by SidechainDiff in Append. D.3.2

We also use the steric clash number metric [1] to evaluate the quality of the side-chain conformations generated. Even if Attnpacker adds additional steric clash loss, DiffAffinity also achieves SOTA results in this metric across all representative methods.

__Q6: Difference with existing torsion angle diffusion model__

In this paper [4], both methods view the torus space $\mathbb{T}^4 \cong [0,2\pi)^4$ and set the exponential map on the torus as $\mathrm{exp}_{x}(y) = x + y\ \mathrm{mod}\ 2\pi$.

In contrast, our model views the torus space $\mathbb{T}^4 \cong (\mathbb{S}^1)^4$ and sets the exponential map on the unit circle $\mathbb{S}^1$ as $\mathrm{exp}_{\mathbf{\mu}}(\mathbf{v}) = \cos(\Vert \mathbf{v} \Vert)\mathbf{\mu}+\sin(\Vert \mathbf{v} \Vert)\frac{\mathbf{v}}{\Vert \mathbf{v} \Vert}$, where $\mathbf{\mu} \in \mathbb{S}^1$ and $\mathbf{v} \in \mathbb{R}^1$. To handle the 4-dimensional torus space $\mathbb{T}^4$, we utilize projection mapping and exponential mapping on the unit circles of each dimension. By applying these mappings to each dimension and taking the Cartesian product of the results, we obtain the exponential mapping $\mathrm{Exp}$ and projection mapping $\mathrm{Proj}$ on the 4-dimensional torus space $\mathbb{T}^4$.

The perturbation kernel is impossible to get an exact solution but requires numeric approximation. The perturbation kernel $p_{t|0}$ in the torus space $\mathbb{T}^4 \cong [0,2\pi)^4$ is proposed as:

$p_{t|0}(x'|x) \propto \sum_{d\in \mathbb{Z}^m}\mathrm{exp}(-\frac{\Vert x-x' + 2\pi d\Vert^2}{2\sigma^2(t)})$.

While the two exponential maps are equivalent to Riemannian exponential maps on the torus and the diffusion process remains the same in the 4-dimensional torus space $\mathbb{T}^4 \cong (\mathbb{S}^1)^4$, using the perturbation kernel $p_{t|0}$ in our paper requires complex derivation based on our specific exponential map. With implicit score matching loss, we are not required to approximate the perturbation kernel numerically.

__Reference__

[1] Matthew et al. An end-to-end deep learning method for protein side-chain packing and inverse folding. Proceedings of the National Academy of Sciences (2023)

[2] Mikita, et al. DLPacker: deep learning for prediction of amino acid side chain conformations in proteins. Proteins: Structure, Function, and Bioinformatics  (2022)

[3] Lin et al. Evolutionary-scale prediction of atomic-level protein structure with a language model. Science (2023)

[4]  Bowen et al. Torsional diffusion for molecular conformer generation. NeurIPS (2022)

---

### Decision · Program_Chairs · 2023-09-21

**Decision:**

Accept (poster)

**Comment:**

This paper proposed a new diffusion model called SidechainDiff to learn the conformational distributions of protein side chains. The diffusion models also learns a contextual representation of mutations on the ppi-interface. Experimental results on existing benchmarks prove the effectiveness of the proposed approach.


The reviewers in general like the proposed diffusion model, which is new and original. The paper would benefit from (1) more discussion on existing methods on side-chain packing and comparison in the experiments; (2) Testing on more antigen than the SARS-COV-2 target.